# Performance in even a simple perceptual task depends on mouse secondary visual areas

Hannah C Goldbach[†], Bradley Akitake[†], Caitlin E Leedy, Mark H Histed*

Unit on Neural Computation and Behavior, National Institute of Mental Health Intramural Program, National Institutes of Health, Bethesda, United States

**Abstract** Primary visual cortex (V1) in the mouse projects to numerous brain areas, including several secondary visual areas, frontal cortex, and basal ganglia. While it has been demonstrated that optogenetic silencing of V1 strongly impairs visually guided behavior, it is not known which downstream areas are required for visual behaviors. Here we trained mice to perform a contrast-increment change detection task, for which substantial stimulus information is present in V1. Optogenetic silencing of visual responses in secondary visual areas revealed that their activity is required for even this simple visual task. In vivo electrophysiology showed that, although inhibiting secondary visual areas could produce some feedback effects in V1, the principal effect was profound suppression at the location of the optogenetic light. The results show that pathways through secondary visual areas are necessary for even simple visual behaviors.

*For correspondence:
mark.histed@nih.gov

†These authors contributed equally to this work

Competing interests: The authors declare that no competing interests exist.

## Introduction

Several kinds of neural computations are required to perform sensorimotor tasks. For example, evidence suggests that some cerebral cortical neurons represent aspects of the sensory world in neural activity, while other neurons decode that representation into a form useful for motor action. An important step in understanding brain function is determining where representations are present, and where computations like decoding occur. These questions depend on knowing which areas are involved during sensorimotor behaviors. Here, we study these questions in the visual system, examining whether secondary visual areas are involved in a simple visual behavior, or whether the behavior can be performed by operating on ('decoding') sensory representations in V1 or other areas directly, bypassing secondary areas.

The rodent visual system includes several different cortical visual areas, which each respond to visual stimuli but emphasize different aspects of representation. The borders and locations of mouse visual areas were initially identified through reversals in retinotopy (*Dräger, 1975*; *Wagor et al., 1980*) and cytoarchitecture (*Caviness, 1975*). Later, injections of anterograde tracers into V1 showed that V1 projects to several secondary (higher) visual areas (*Wang and Burkhalter, 2007*). More recently, the boundaries of cortical visual areas in mice have been examined through mapping retinotopy via intrinsic imaging (*Schuett et al., 2002*; *Kalatsky and Stryker, 2003*; *Garrett et al., 2014*; *Juavinett et al., 2017*) and calcium imaging (*Andermann et al., 2011*; *Murakami et al., 2017*; *Zhuang et al., 2017*). These imaging studies have identified as many as 16 individual, retinotopically mapped areas (*Zhuang et al., 2017*) that vary in the strength of their connectivity to V1. Those with the largest proportions of projections from V1 are also the most heavily studied: the lateral and medial output pathway areas, including LM (lateromedial), AL (anterolateral), RL (retrolateral), and PM (posteriomedial) (*Froudarakis et al., 2019*). While more than half of V1 projections target these four secondary visual areas (*Han et al., 2018*), V1 also projects to a number of other brain

areas including frontal cortex and basal ganglia (*Han et al., 2018*; *Khibnik et al., 2014*), both of which are involved in decision-making and action selection.

We sought to determine whether activity in secondary visual areas was essential for visual perceptual behaviors in mice. Alternatively, it could be that such activity is not always used, and for some tasks, information flows out of V1 via projections that bypass secondary areas, such as V1's projections to frontal cortex or basal ganglia. As an example, consider an analogous question in a larger species. In the primate visual object ('what') pathway, V1 carries representations of small visual edges (*Hubel and Wiesel, 1968*), while areas in inferotemporal (IT) cortex represent complex objects such as faces (*Tsao, 2014*). Identifying the orientation of contours and edges could be carried out by a neural computation that reads out, or decodes, V1 directly, bypassing secondary visual areas such as those in IT cortex. In that case, suppressing IT cortical areas without affecting V1 would impact tasks that require face perception, but leave tasks that involve orientation discrimination unaffected. On the other hand, some data, for example, on unconscious versus conscious perception (*Leopold, 2012*), support the idea that macaque V1 is simply a conduit for information to secondary visual areas, with no behaviorally relevant direct readout. In the mouse, we are able to investigate which cortical areas are involved in a given behavior using transgenic lines expressing optogenetic proteins. Thus, we can inactivate several different cortical visual areas just by moving the light stimulus. Using this approach, we studied the extent to which secondary visual areas contribute to mouse perceptual behavior using a simple visual contrast-change detection task. Animals were trained to report changes in visual stimulus contrast (detecting the appearance of stimuli on a uniform background) for which V1 contains substantial information. Results obtained in this paradigm would shed light on the interaction of visual cortical regions: if secondary visual areas are involved in processing the mere appearance of a stimulus, perhaps the simplest visual perceptual task, it seems likely that a large fraction of visual perceptual tasks also rely on both V1 and secondary areas.

Neural activity in V1 carries substantial information about visual stimulus contrast (*Albrecht and Hamilton, 1982*; *Sclar et al., 1990*; *Vaiceliunaite et al., 2013*). Neurons in mouse V1 are tuned for oriented visual stimuli, with a tuning curve that peaks at the cell's preferred orientation. The same neurons simultaneously encode contrast: as stimulus contrast increases, single V1 neurons increase their firing rates monotonically (*Busse et al., 2011*; *Glickfeld et al., 2013a*; *Khastkhodaei et al., 2016*). As expected from single neurons' responses, the average, or summed, V1 response does not completely saturate, even up to 100% contrast (*Glickfeld et al., 2013a*; *Histed, 2018*). These features suggest that mouse V1 activity, by itself, can be simply (linearly) decoded to support contrast-change detection. While the structure of population neural variability could, in principle, make linear decoding difficult, linear decoders that do not take trial-to-trial variability into account perform well on decoding orientation from mouse V1 (*Berens et al., 2012*; *Rumyantsev et al., 2020*; *Stringer et al., 2019*; *Kafashan et al., 2021*). This suggests stimulus contrast could be extracted directly from the V1 population by a simple (linear) decoder without requiring additional transformation from areas beyond V1. Thus, secondary visual areas might not be involved in the task at all, in which case inhibiting them would not impact animals' behavior.

To test this idea, we trained mice on a contrast-change detection task in which animals report the presence of a flashed Gabor by releasing a lever. This task was previously found to require V1; suppressing V1 optogenetically produced retinotopically-specific impairments in contrast-detection performance (*Glickfeld et al., 2013b*; *Jin and Glickfeld, 2020*). This impairment was substantial but not complete, so while other visual pathways, such as those involving the superior colliculus (SC) may also contribute to this behavioral task, performance depends on cortical processing in V1. We first confirmed that V1 contributes in our task using the VGAT-ChR2-EFYP transgenic mouse line (*Zhao et al., 2011*) to optogenetically stimulate all inhibitory neural classes. We found that inhibiting V1 in this way negatively affects perceptual performance. We then inhibited secondary visual areas and found that suppressing lateral areas (e.g., LM/AL and RL) and medial areas (PM) also negatively affects performance. Via electrophysiology and widefield calcium imaging, we found that inhibiting secondary visual areas primarily acts to suppress secondary area visual responses, rather than suppressing V1 responses via feedback. These data show that secondary visual areas are involved in a task that, in principle, could have been performed by direct V1 readout alone. We also found that the effect size caused by suppressing secondary areas can be large, comparable to what we see when suppressing V1 using the same optogenetic light intensity. In contrast, suppressing PM, the largest medial area, only weakly affected performance. Paralleling these behavioral findings, we

found that any feedback effects on V1 neural activity were negligible when PM was suppressed, whereas inhibiting LM produced some feedback suppression of V1 at high light intensity, arguing that activity in LM has some feedback effect on V1 even though the feedback effects cannot alone explain the changes in perception. Finally, in all areas the behavioral effects are primarily due to changes in sensitivity ($d'$) and not false alarm (FA) rates. In sum, lateral secondary visual areas (LM/AL and RL) form an interconnected network with V1 that contributes to performance in even simple, visually guided behaviors, for which substantial information is available within, and linearly decodable from, V1.

## Results

To understand the role of secondary visual areas, we trained mice to perform a contrast-change detection task in which psychophysical measurements are possible during optogenetic inhibition. We first describe the details of the behavioral and optogenetic paradigm.

We used transgenic mice of both sexes that expressed ChR2 in all inhibitory neurons (VGAT-ChR2-EYFP mouse line, *Zhao et al., 2011*) and implanted a cranial window over occipital cortex. Because this mouse line expresses a yellow fluorophore, which can make it difficult to perform some kinds of calcium imaging, we found the locations of areas in visual cortex using hemodynamic intrinsic-signal imaging. We identified a series of retinotopic locations within V1 (Materials and methods, *Figure 1*, *Figure 1—figure supplement 1*), and based on the location of V1, we identified several secondary visual areas, areas AL, LM, RL, and PM (among the secondary visual areas whose locations are most reliable, *Garrett et al., 2014*). We cemented an optical fiber over the cranial window to create a light spot targeted to the area of visual cortex we wished to inhibit, and trained these animals on a contrast-increment change detection task (Materials and methods, *Figure 1A*). We measured with a camera the intensity profile on the surface of the brain (approximately a two-dimensional Gaussian due to the Gaussian radial profile of light leaving the fiber), and found the 50% contour of the light intensity, which we use to show the boundaries of each spot (Materials and methods; *Figure 1—figure supplement 2*). Behaving animals were head-fixed in front of a display monitor and trained, for a liquid reward, to release a lever in response to the onset (increase in contrast, from a neutral gray background) of a small Gabor patch (full-width at half-maximum [FWHM] = 14° of visual angle). We varied the contrast-change magnitude to obtain psychometric curves, which measure the contrast needed for animals to perform the task at a set percentage correct, that is their perceptual threshold (*Figure 1C*; below we show that behavioral effects are similar for percent correct and for perceptual sensitivity, or $d'$).

Once mice were trained to threshold, we introduced a pulsed blue LED train (470 nm; on for 200 ms, off for 1000 ms) throughout each trial. On 50% of trials (ON trials), the LED pulse occurred with the visual stimulus. Due to latency between visual stimulus onset and first spikes in the cortex, we turned the light pulse on 55 ms after visual stimulus onset. We found that this timing produced large changes in perceptual performance (*Figure 2A,B*), consistent with physiological recordings, where we found that this light onset time was 8–12 ms before the visual response occurred in V1 (*Figure 6—figure supplement 1*). On the remaining trials (OFF trials), the visual stimulus appeared while the light was off. The timing of our inhibitory light pulses corresponds well to *Resulaj et al., 2018*, which reports that effects on behavior, in response to optogenetic inhibition, begin between 40 and 80 ms after visual stimulus onset.

To limit the information the animals could gain about correct response (i.e., visual stimulus onset time) from the optogenetic light pulses, the light pulse train was present on each trial and was randomized by varying the first pulse onset time after start of the trial (Materials and methods). On each trial, the visual stimulus onset time was chosen randomly (according to a geometric distribution), the light pulse train phase offset was chosen randomly (uniform distribution), and the visual stimulus onset time was adjusted downward to occur with the previous light pulse (ON trials) or after the previous light pulse (OFF trials). Thus, each trial, whether ON or OFF, contained a similar pulse train. Finally, we randomly intermixed ON and OFF trials. Together, these factors made it difficult for animals to use information about the light pulses to identify the type of trial and change their strategy (such as their perceptual criterion) based on trial type.

To verify that animals were not able to use light pulse information to guide their responses, we performed several controls and control analyses. We found (details below): (1) the optogenetic

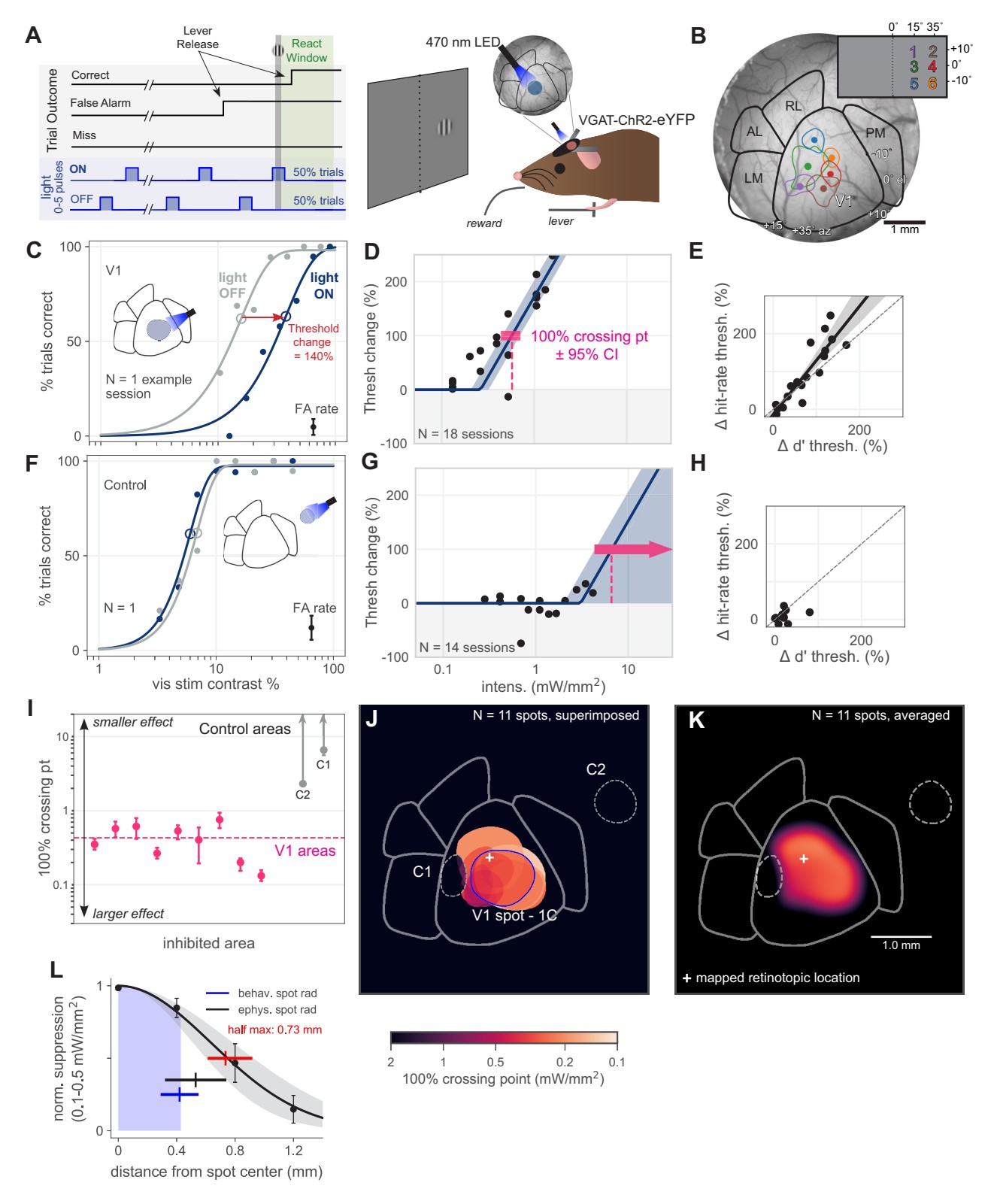

**Figure 1.** Optogenetic inhibition of V1 confirmed to affect behavior. (**A**) Schematic of the visual detection behavior task. Animals release a lever when they detect the onset of a visual stimulus. Stimuli were Gabor patches of varying contrast presented on a neutral gray screen. Cortical activity was suppressed via optogenetic activation of inhibitory neurons (VGAT-ChR2-EYFP mouse line). On each trial, a train of light pulses was applied to the cortex (Materials and methods). Visual stimuli were delivered either during a light pulse (light ON trials) or when the light pulse was off (light OFF trials).
*Figure 1 continued on next page*

*Figure 1 continued*

(B) Cortical areas were identified using hemodynamic intrinsic imaging of cortical responses to a set of visual stimuli at different positions within visual space (Materials and methods, *Figure 1—figure supplement 1*). (C) LED stimulation in V1 decreases performance (single session; light intensity = 0.46 mW/mm$^2$; rightward shift indicates decreased performance). Filled circles: performance at one contrast. Open circles: threshold (63% point of fit curve). False alarm (FA) rate is estimated FA hazard rate (Materials and methods). Lower asymptote of psychometric function is fixed at zero. Upper asymptote (lapse rate) is allowed to vary. (D) Piecewise-linear function fit to all sessions from the spot in panel C, measuring the intensity required to double the psychometric threshold (N = 18 sessions, 100% crossing point, pink; mean 0.57, bootstrap 95% CI 0.43–0.71 mW/mm$^2$). Slopes were fit by pooling across stimulation locations and experimental sessions; fitting slope separately for each session produced similar results (*Figure 1—figure supplement 3*). (E) Change in $d'$ threshold increase vs change in hit-rate threshold increase, for the spot in panel C. (F) Single session example behavior from LED stimulation in a control area outside of V1 (light intensity = 1.3 mW/mm$^2$), same conventions as panel C. (G) Same as D, fit to sessions from control spot in panel F (N = 14 sessions, mean 100% crossing point 6.6, bootstrap 95% CI 4.3–7.6 mW/mm$^2$). (H) Same as panel E, fit to sessions from the control spot in panel F. (I) 100% crossing points for all V1 spots and two control spots, as determined by regression fits in E and G. Ordering of spots is arbitrary; points are offset along x axis to allow display of confidence intervals for each point. (J) Contours of all V1 spots and controls, color weighted by 100% crossing point. Spot contours are the full width at half-max of the Gaussian light distribution produced by the fibers (N = 11 spots, *Figure 1—figure supplement 2*). (K) Heatmap of effect size in V1, generated by averaging the 100% crossing point for each spot at each pixel (N = 11 spots). (L) Spatial fall-off of inhibition. Y-axis: suppression of visual responses measured with electrophysiology (silicon probe, four shanks), light intensity between 0.1 and 0.5 mW/mm$^2$ (N = 4 experiments, black line: Gaussian fit, gray: 95% CI via bootstrap; data error bars: SEM, N = 45 single units). Full width at half max of Gaussian fit = 0.73 mm, bootstrap 95% CI 0.61–0.92 (red line). Electrophysiology spot radius: 0.53 ± 0.21 mm (black line, mean ± SD). Behavioral spot radius: 0.42 ± 0.13 mm (blue shaded area and line, mean ± SD, N = 34 inhibitory light spots). Since behavioral spots are smaller on average than what was used for these physiology experiments, the spatial effect of inhibition during behavior is likely more restricted than shown by these suppression data (black curve, red bar).

The online version of this article includes the following figure supplement(s) for figure 1:

**Figure supplement 1.** Visual area identification via imaging.
**Figure supplement 2.** Light spot size calculations.
**Figure supplement 3.** Slope determination for piecewise-linear function.
**Figure supplement 4.** Lapse rate, estimated false alarm (FA) hazard rate, and psychometric function slope vary little with stimulation.

stimulation systematically *decreased* animals' performance, (2) false alarm hazard rate was fairly constant through the optogenetic pulse period and did not vary between trial types, (3) stimulation at some control locations did not change performance, and (4) sensitivity ($d'$) analyses yielded qualitatively the same results as hit-rate measures.

Finally, though cognitive factors like attention and motivation can change over experimental sessions from one day to the next, we expect such factors to act similarly on both trial types. To control for day-to-day changes in attention, motivation, or perceptual criterion choice (*Macmillan and Creelman, 2004*), we always compare ON and OFF trials within each daily experimental session (e.g., gray vs blue curves, *Figure 1C,F*).

In sum, this change detection behavioral paradigm allows us to examine effects on visual perception due to inhibition of different mouse visual areas.

## V1 inhibition confirmed to impair behavior

We first sought to validate our methods in V1. Based on previous work (*Glickfeld et al., 2013b*), we expected that inhibiting V1 during a contrast-change detection task would impair the animals' behavior. Indeed, we found that inhibiting V1 during behavior, via a retinotopically aligned spot of light over V1, produced an intensity-dependent impairment (increase in psychometric threshold; *Figure 1C*). In order to obtain a measure of how effective the optogenetic inhibition was at impairing perception, for each stimulation light spot we fit a piecewise-linear function to the threshold increases as a function of optogenetic light intensity. This yielded an estimate of the intensity required to produce a 100% increase in threshold (yielded the 100% crossing point; *Figure 1D*, *Figure 1—figure supplement 3*) and allowed us to compare the behavioral impact of inhibition across light spots (*Figure 1I–K*).

In principle, optogenetic inhibition could have merely distracted the animal from the task instead of acting on visual perceptual processes. If so, the change in attentional state should produce errors on easy trials with high contrasts – that is, the lapse rate should increase. To avoid this, animals were

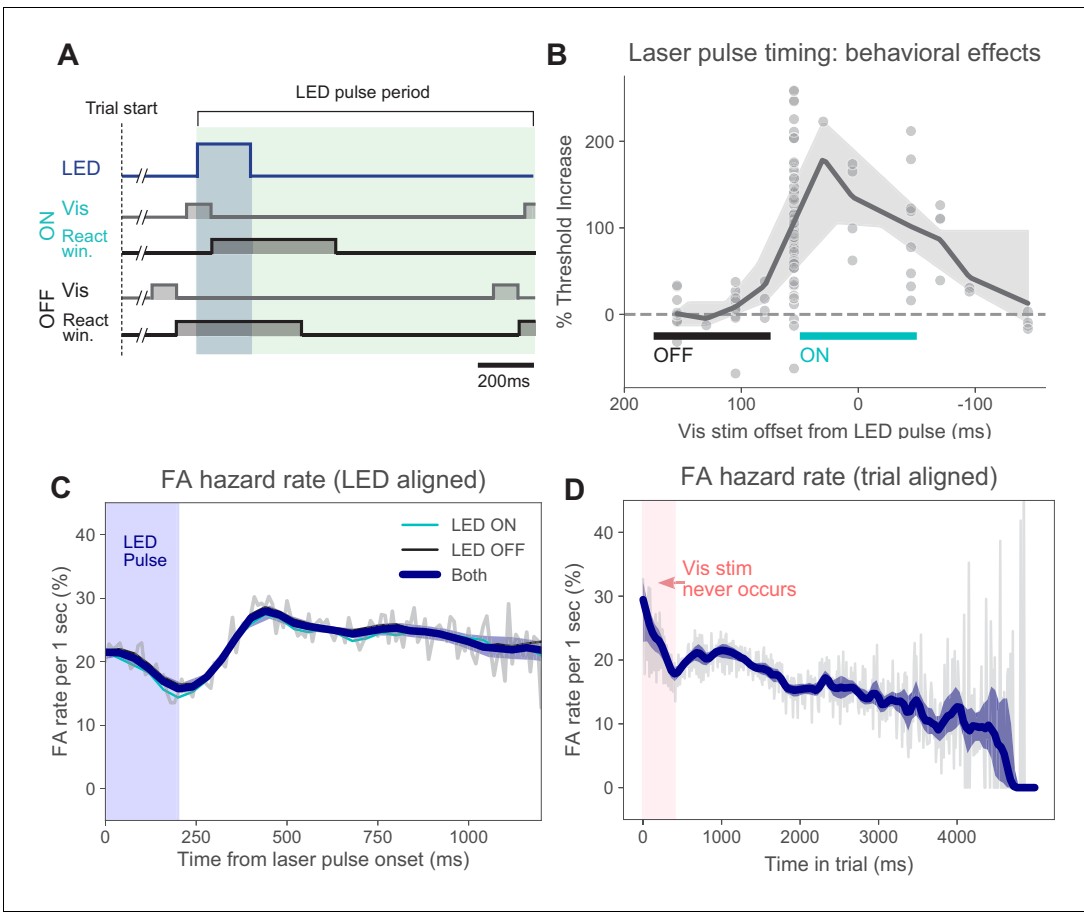

**Figure 2.** Timing of stimulus relative to pulses and false alarm rates. (**A**) Timing of task events within LED pulse period. LED pulse trains were the same on each trial except the onset phase was randomized (uniform distribution, Materials and methods). Light pulse duration 200 ms (dark blue shaded region). Off duration 1000 ms (green shaded region), therefore, pulse period is 1200 ms. Pulse onset timing was chosen based on measured effects on perceptual behavior with ON pulses (panel **B**). Trials could last for several pulse periods. (**B**) Threshold increase (%) across visual stimulus offsets. Points indicate changes in perceptual threshold in individual experiments where timing offset between stimulus onset and LED pulse onset was fixed (N = 112 sessions from N = 5 animals). Solid line: LOWESS fit. (**C**) False alarm hazard function (probability of false alarm given the trial has lasted until that time; calculated over N = 50,382 trials that resulted in false alarms, pooled across animals and light intensities). Since stimulus onset times are randomized in this task, the false alarm hazard function is the measure that parallels hit rates for signal detection theory-based analysis (*Macmillan and Creelman, 2004*). Here false alarm hazard rates are plotted relative to LED pulse onset time (which is randomized relative to trial onset time). Thin gray line: false alarm hazard computed in 1 ms bins (all hazard rates are normalized to units of false alarm rate per 1 s interval: y-axis). Thick blue line: LOWESS fit. Shaded region (mostly hidden): 95% CI via bootstrap. Thin lines (also largely hidden): false alarm hazard for ON and OFF trials, showing ON and OFF trial types have similar false alarm hazard rates. (**D**) False alarm hazard function plotted relative to trial start time (same trials as **C**). Visual stimulus onset time and LED pulse onset time are both randomized relative to trial start in each trial (Materials and methods). While each trial begins with a tone at time 0, the visual stimulus onset never occurs in the first 400 ms; the initial increase in false alarm rates likely reflects subjects' slight failure to inhibit responses to trial start. For longer times, fewer trials are in the data set and estimates become more variable. Error bars: 95% CI via bootstrap.

trained to have low lapse rates. Additionally, we found that lapse rates were unchanged by optogenetic inhibition (*Figure 1—figure supplement 4*).

Also, if optogenetic inhibition was producing changes in cognitive state instead of acting on visual processing, we might expect changes in the slope of the psychometric function with

optogenetic inhibition. We examined psychometric slopes and found they did not vary with optogenetic inhibition (*Figure 1—figure supplement 4*). Thus, optogenetic inhibition shifted perceptual threshold without changing slope. In a signal detection theory framework, this means that our inhibition changed the amount of visual information available without changing information variability or noise (e.g. *Macmillan and Creelman, 2004*). Note that variation in lapse rate or slope have similar implications in detection tasks, as we use here, and for 2-alternative forced choice (2AFC) tasks (*Macmillan and Creelman, 2004*; *Murasugi et al., 1993*). However, one principal difference between the two types of tasks is that detection tasks allow subjects more freedom to vary their internal perceptual criterion. Because criterion changes are a function of both hit rate and false alarm (FA) rate, we next analyze false alarm rates.

Our task design tends to discourage animals from changing their false alarm rates. Signal detection theory characterizes changes in optimal performance via sensitivity ($d'$), separate from changes in internal perceptual criterion or willingness to respond. For example, in our detection task, if subjects merely release the lever more often to a perceptual stimulus, with no change in the stimulus and no added optogenetic stimulation, their percent correct will increase, however, so will their false alarm rate. In our task, the stimulus onset time is chosen randomly up to a long maximum onset (several seconds; Materials and methods). Because of the random stimulus onset time, the relevant signal detection measure for false alarms is the false alarm hazard rate (*Papoulis and Pillai, 2002*): the probability that a false alarm will happen in the next instant of time, given that the trial has lasted until that time without a false alarm or a stimulus occurring. And in a temporally-extended task like this, small changes in false alarm hazard rate translate into large changes in the fraction of trials ending in a false alarm, and thus trials on which reward is withheld. Therefore, in our task, relatively small changes in subjects' moment-to-moment false alarm rate lead to large changes in reward, incentivizing the subjects to choose a relatively stable false alarm rate.

Matching this expectation, we found that the false alarm hazard rate was changed little by several behavioral variables. First, the hazard rate was similar for ON and OFF trials (*Figure 2C*), across all intensities tested (*Figure 1—figure supplement 4*). When we plot the false alarm hazard rate time-locked to the laser pulse train, we find a small decrease in the average false alarm hazard rate with pulse onset, as might be expected from optogenetic inhibition. We also found that the false alarm hazard rate declined slightly over time in trial (*Figure 2D*). If subjects were on average using a temporal guessing strategy, we would expect to see a peak in the false alarm hazard rate at a particular moment in time, but such a peak was not present. A second type of temporal guessing strategy ruled out by these data is one in which animals guess (false alarm) more during longer trials. For distributions of stimulus onset times that are not perfectly geometric (discrete exponential, Materials and methods), stimulus onset becomes more likely as trials last longer, which can lead to elevated false alarm rates. However, again this is not what we observe — false alarm rates do not go up as trials get longer. In sum, our data indicate that animals, on average, performed the task accurately without using timing-based guessing strategies.

Since false alarm hazard rates changed only a small amount, we expected that effects measured with sensitivity ($d'$) would be similar to those measured with hit rates. To test this, we computed $d'$ thresholds in an analogous manner to hit rate thresholds. We found the estimated false alarm hazard rate for each behavioral session by taking the fraction of each type of trial (ON or OFF) terminated by a false alarm and dividing by the average trial length for that trial type. (Computing the full false alarm hazard rate for each session was impractical due to the large number of trials needed to estimate hazard rates *Figure 2C,D*: N = 50,382 trials). Using the resulting estimated rate, we computed $d'$ for each session and fit a Naka–Rushton function to $d'$ values at each contrast level (Materials and methods, *Herrmann et al., 2012*). We did this for light ON and light OFF trials to find two thresholds for $d'$. Sensitivity threshold changes were strongly correlated with hit rate threshold changes (*Figure 1E*). We also find that the patterns of effects with inhibition of different visual areas are similar for hit rate and for sensitivity. Because there was little change in false alarm rate, and thus $d'$, as expected, thresholds measured with sensitivity and with hit rate were related (for all spots in V1, Wilcoxon signed-rank test W = 15, $N_1 = N_2 = 9$, p = 0.37), here and below we primarily calculate threshold changes on hit rate.

Inhibiting control areas, away from visual areas, should produce little effect on visual task performance. Indeed, we found inhibiting control areas produced no effect on threshold (*Figure 1F*). While the light OFF threshold in panel F varies from the light OFF threshold in panel C, this was not a large

effect across our data (light OFF threshold for all V1 sites: mean 12.5%, 95% CI 11.3–13.7%; for all control sites, mean 8.7%, 95% CI 7.7–9.8%). Moreover, there was no systematic relationship between threshold change and light OFF threshold (across all V1 sites, OLS regression, pct change vs. contrast, slope n.s. different from zero: 95% CI -3.8 - 1.1, N = 103 sessions, p = 0.29). To determine the minimum intensity necessary to produce a 100% increase in threshold (*Figure 1G*), we fit the same piecewise-linear function (*Figure 1—figure supplement 3* shows how slope was fit) to these data across all collected intensities. For these control light spots, the data put a lower bound on the intensity required to produce a behavioral effect. In contrast, the upper confidence interval for these plots extends upward indefinitely, as we found no intensity high enough for these light spots to cause behavioral effects (that is, no upper bound). As with V1 inhibition, changes in false alarm rates were small (summarized in *Figure 1—figure supplement 4*). With both hit rate and false alarm rate showing only small changes, sensitivity (*d'*) changes were also small (*Figure 1H*).

To measure the spatial falloff of optogenetic inhibition, we used electrophysiological recordings from cortical layer 2/3 neurons (up to 500 μm below the brain surface, see Materials and methods) combined with optogenetic stimulation (*Figure 1L*). We measured suppression at powers used to produce behavioral effects (around the 100% crossing points seen in V1; 0.1–0.5 mW/mm$^2$; N = 4 recording sessions). The half-max radius of suppression (0.73 mm, 95% CI 0.61–0.92) was slightly larger than the half-max radius of the light spots used in the recordings (0.53 ± 0.21 mm, mean ± SD).

In these electrophysiological measurements (*Figure 1L*), we characterized the spatial falloff of suppression of visual responses, because cortical baseline (spontaneous) firing rates are more sensitive to optogenetic manipulation than evoked visual firing rate changes are (*Glickfeld et al., 2013b*; *Histed, 2018*). We positioned an array of four electrodes, so three electrodes were in V1 and the last electrode shank was in PM, which was targeted with optogenetic light. In this configuration, the farthest electrode from PM (1.2 mm away), which shows the smallest suppression, is at the center of the V1 retinotopic representation for the stimulus. As discussed below, there is little feedback effect on V1 of stimulating PM (and, in fact, at this light intensity, little feedback effect of stimulating any secondary visual area). Recordings spanning from LM to V1 (instead of from PM to V1, as here, *Figure 1L*) show increased feedback suppression in V1, but overall similar spatial falloff of suppression (*Figure 6—figure supplement 2*, compare 50% points in *C*, yellow). The specificity of our optogenetic manipulations is also confirmed by behavioral measurements (V1, above; other areas, below), as the pattern of effects on perceptual behavior is dependent on where the light is delivered to the cortex.

Our V1 results were consistent across animals and across V1 light locations (mean 100% crossing point 0.43, bootstrap 95% CI 0.30–0.55 mW/mm$^2$; *Figure 1I*). In total, we tested nine spots within V1, all of which produced substantially larger effects (i.e. lower 100% crossing points) than did control spot inhibition (*Figure 1J*; N = 5 light spots, mean control spot 100% crossing point 4.3, bootstrap 95% CI 2.0–6.9 mW/mm$^2$; Mann–Whitney U = 0.0, V1 N = 9 light spots, control N = 5 light spots, p = 0.0017 two-tailed). Notably, inhibition of an area placed within V1 but offset from the retinotopic location of the visual stimulus produced no effect (*Figure 1J*, control spot 1). To visualize the effects of inhibiting V1 across our stimulation locations, we averaged the 100% crossing points at each pixel within V1 to generate a heatmap (*Figure 1K*). Because this heatmap representation is influenced by the locations of the optogenetic light spots, it is not intended to be fully quantitative, but instead to provide a visual guide to assess the spatial pattern of the effects given the spot locations used.

Together, these data confirm that inhibiting mouse V1 negatively affects an animal's ability to perform a simple contrast-change detection behavior.

## Inhibiting areas lateral to V1 (LM/AL and RL) degrades contrast-change detection

We next inhibited secondary visual areas, beginning lateral to V1 in LM, AL, and RL. From here on, we refer to LM and AL together as LM/AL, as our optogenetic light spots could not easily differentiate between the two. AL is small relative to the spatial spread of our inhibition (*Figure 1L*) and further, the retinotopic representation of our stimulus is on the border between these two areas (*Garrett et al., 2014*; *Zhuang et al., 2017*).

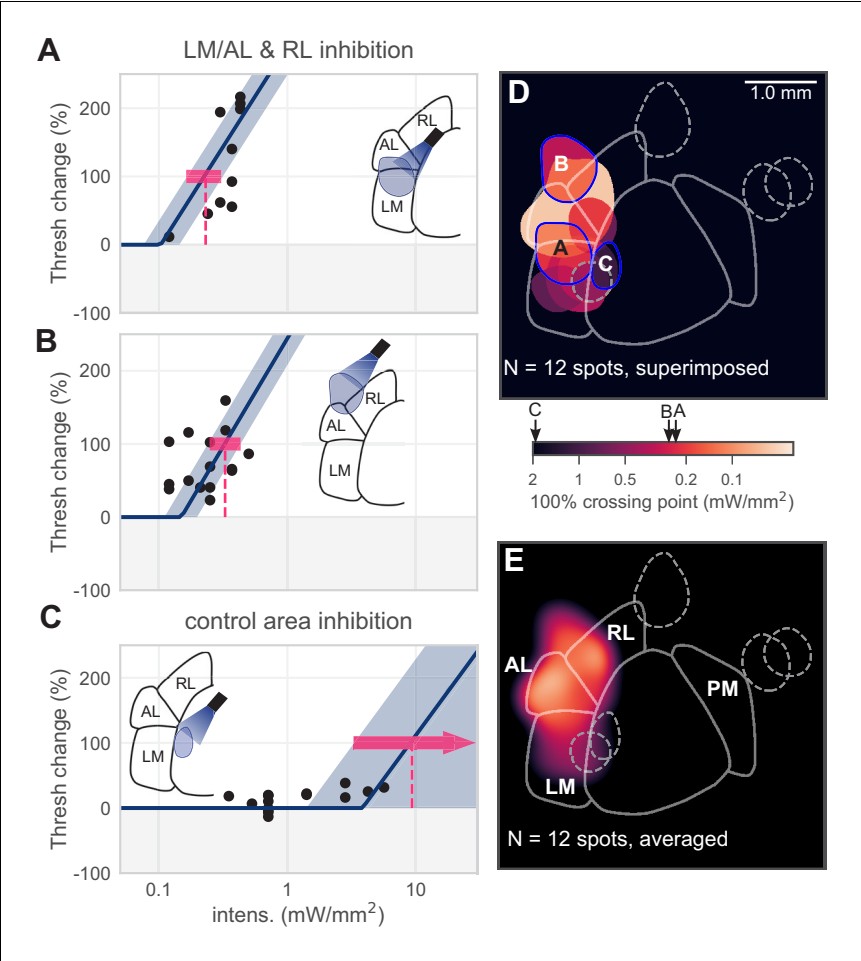

**Figure 3.** Inhibiting lateral areas degrades contrast-change detection behavior. (**A**) All sessions from an example spot on the border of AL and LM, showing a large effect (N = 12 sessions, mean 100% crossing pt. 0.23, bootstrap 95% CI 0.16–0.31 mW/mm²). (**B**) All sessions from an example spot within RL, showing a large effect (N = 18 sessions, mean 100% crossing pt. 0.33, bootstrap 95% CI 0.25–0.43 mW/mm²). (**C**) All sessions from a control area between LM and V1, showing no effect even at high intensities (N = 15 sessions, mean 100% crossing pt. 8.7, bootstrap 95% CI 4.6–9.3 mW/mm²). (**D**) Map of spots within LM/AL/RL and control locations, colored by 100% crossing pt. Crossing points from panels **A**–**C** are shown on color bar (black arrows). (Note slight difference for visual display in color bar extents compared to *Figure 1J,K*). (**E**) Heatmap of effect size, generated by averaging the 100% crossing point at each pixel; pixels with no data are colored black.
The online version of this article includes the following figure supplement(s) for figure 3:

**Figure supplement 1.** Control spots between V1 and LM.

Inhibiting LM/AL and RL produced large increases in psychometric threshold at low intensities, resulting in small 100% crossing points (mean 100% crossing point 0.37, bootstrap 95% CI 0.20–0.56 mW/mm²) (*Figure 3A,B*). Inhibiting at a control spot between V1 and LM (same as control spot 1, *Figure 1*) produced no effect on behavior, even at high intensities (N = 15 sessions, *Figure 3C*). The difference between the behavioral effects of inhibiting lateral areas versus control areas was large (Mann–Whitney U=0.0, lateral N = 7 light spots, control N = 5 light spots, p = 0.0029 two-tailed; *Figure 3E*).

A potential concern is that optogenetic inhibition of LM/AL might merely spread across the cortex to the V1 representation and exert direct effects on V1 to create behavioral changes. In addition to our electrophysiological characterizations (*Figure 1L* and other imaging and physiology data below), we ruled out this possibility in two ways. First, we examined the behavioral effect size based

on distance from the V1 representation (that is, the location in the V1 retinotopic map that produces the largest response to our stimulus location) (*Figure 3—figure supplement 1A*). We found no relationship between distance from V1 and threshold shift for control spots or secondary visual area targets. Second, as a within-animal control (N = 2 animals), we moved the light spot from the V1 retinotopic representation location to an intermediate location between V1 and LM/AL. In these experiments we found a dramatic drop-off in the size of the behavioral effect (*Figure 3—figure supplement 1B–F*). Together, these data indicate that the behavioral effects were not due to inhibition spread across the cortex to V1, but were instead due to optogenetic inhibition of the secondary areas themselves.

The piecewise-linear function we used fixed the slope of each curve. While there was some variability in slope across data points (*Figure 3A,B*), we used this approach because, in V1, allowing slopes to vary did not appreciably change the estimated crossing points (*Figure 1—figure supplement 3*).

We also examined, across cortical areas, how false alarm rate, lapse rate, and psychometric function slope changed with optogenetic inhibition (*Figure 1—figure supplement 4*). We found little evidence that any of these quantities changed across areas (only one pairwise comparison, lateral areas vs. control areas for false alarm rate, was statistically significant). Thus, the primary behavioral change due to optogenetic inhibition was on animals' perceptual threshold: the amount of stimulus contrast needed to perform the task — that is, their perceptual sensitivity.

In sum, inhibiting these lateral secondary visual areas resulted in substantial degradation of the animals' ability to perform the contrast-change detection task.

## Inhibiting PM produces smaller effects than inhibiting lateral areas (LM/AL and RL)

In addition to inhibiting areas within LM, AL, and RL, we also inhibited locations medial to V1, within area PM. We found that, on average, inhibiting spots within PM produced little effect on behavioral threshold (*Figure 4A,B*, N = 5 spots, mean PM 100% crossing point 2.1, bootstrap 95% CI 0.70–3.5 mW/mm$^2$). Inhibiting control areas located outside PM had small or undetectable effects on threshold, even at relatively high light intensities ($\geq$5 mW/mm$^2$, *Figure 4C*). Of five light spots targeted to PM, two showed obvious effects, which were of medium size compared to effects in V1 and the lateral areas (*Figure 4D,E*).

Most PM spots produced small effects, with the one PM spot that produced a moderate effect at the most anterior point of PM. For V1 and the lateral areas, our light spots were matched to those areas' retinotopic maps (*Garrett et al., 2014*; *Zhuang et al., 2017*). However, the anterior area of PM has a retinotopic representation that is slightly more temporal in azimuth than the stimulus we used (25° azimuth). To determine if smaller PM effects relative to LM/AL/RL were due to the retinotopic position of the stimulus, we measured the effect of inhibiting PM while animals performed the same task with stimuli at 45-65° azimuth (*Figure 4—figure supplement 1*). We found small or nonexistent effects with these stimuli (mean threshold changes were near zero: −3.2 ± 17.4%, mean ± SD; linear regress. slope p=0.40). In sum, for the same stimulus used for V1 and LM measurements, and for a more eccentric stimulus that may be better represented in PM, the effects of inhibiting PM were significantly smaller than those that arose from inhibiting LM/AL or RL.

## Inhibiting LM/AL/RL produces similar effects on behavior as inhibiting V1

We compared the strength of behavioral effects of inhibition across areas by computing the mean effect (mean 100% crossing point, the measure of how much light is required to produce a fixed degradation in behavior) in each region: lateral areas, PM, or V1. We found that the mean effect was similarly strong (yielding lower crossing point intensity) in the lateral areas and in V1 (*Figure 5A*; Mann–Whitney U = 26.5, lateral N = 7 spots, V1 N = 9 spots, p = 0.31 two-tailed), while the mean effect across all PM spots was weaker (larger crossing point intensity) than in V1 and in lateral areas (V1 vs medial: Mann–Whitney U = 9.5, V1 N = 9 spots, medial N = 5 spots, p = 0.048 two-tailed; lateral vs medial: Mann–Whitney U = 6.0, lateral N = 7 spots, medial N = 5 spots, p = 0.037 two-tailed). The average response heat maps (*Figure 5B,C*; lighter colors are stronger effects on behavior and

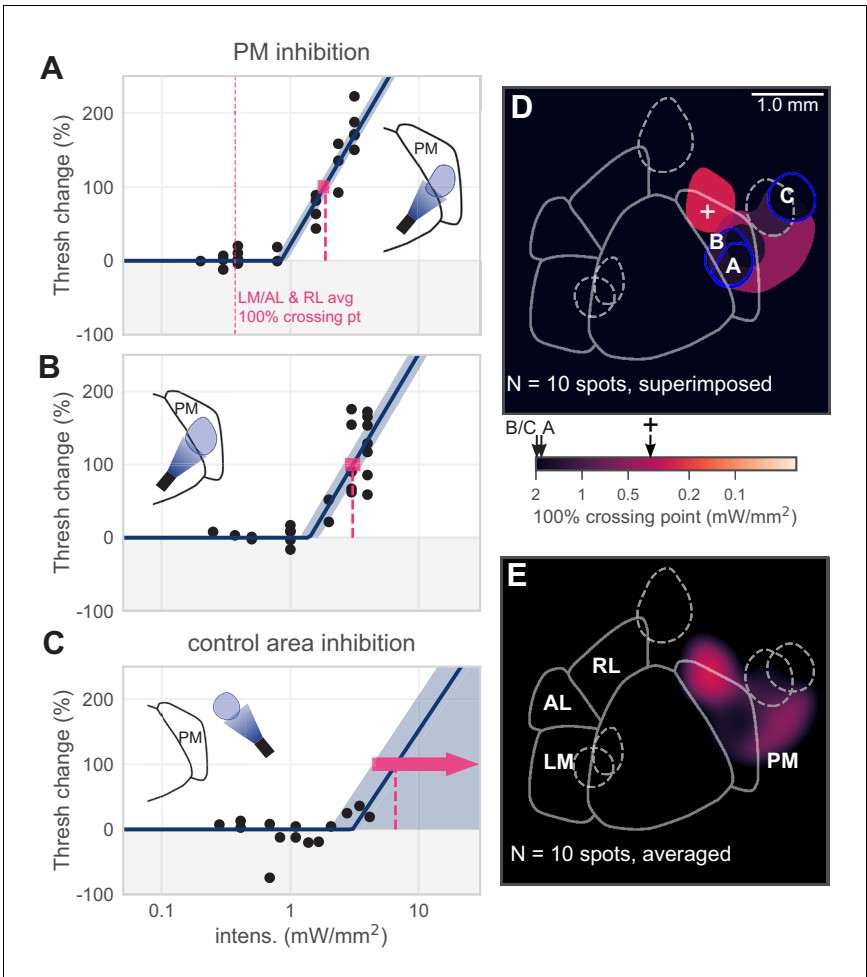

**Figure 4.** Inhibiting medial areas produces weaker effects on behavior. (**A**) All sessions from an example spot within PM, which produced a weak effect (N = 22 sessions, mean 100% crossing pt. 1.9, bootstrap 95% CI 1.6–2.0 mW/mm$^2$). Vertical pink dashed line represents average crossing point from LM/AL and RL (0.37 mW/mm$^2$). (**B**) All sessions from a second example spot within PM, which also produced a weak effect (N = 23 sessions, mean 100% crossing pt. 3.1, bootstrap 95% CI 2.7–3.5 mW/mm$^2$). (**C**) All sessions from a control area outside PM, showing very little effect even at high intensities (N = 14 sessions, mean 100% crossing pt. 6.6, bootstrap 95% CI 4.3–7.6 mW/mm$^2$). (**D**) Map of spots within PM and control locations, colored by 100% crossing point. Crossing points from panels **A–C** are shown on color bar (black arrows). Black cross on color bar denotes average 100% crossing pt. from LM/AL and RL. (**E**) Heatmap of effect size, generated by averaging the 100% crossing point at each pixel. The online version of this article includes the following figure supplement(s) for figure 4:

**Figure supplement 1.** Moving the visual stimulus does not affect PM results.

thus smaller crossing point intensities) show the spatial extent of these effects given the spot positions we tested.

Perceptual behavioral performance can be affected both by changes in sensitivity, the amount of information subjects can use about the stimulus change, or by changes in perceptual criterion, the subjects' willingness to respond 'yes' (in this case, willingness to release the lever). A more permissive criterion increases hit rate at the cost of a higher false alarm rate. In the V1 data (*Figure 1*), we found that optogenetic inhibition caused little change in false alarm rate, and thus sensitivity (*d'*) changes mirrored the hit rate changes. We also examined false alarm rates when inhibiting other areas and found minimal differences in estimated false alarm hazard rates between areas (*Figure 1—figure supplement 4*). Moreover, false alarm hazard rate did not increase with intensity (*Figure 1—*

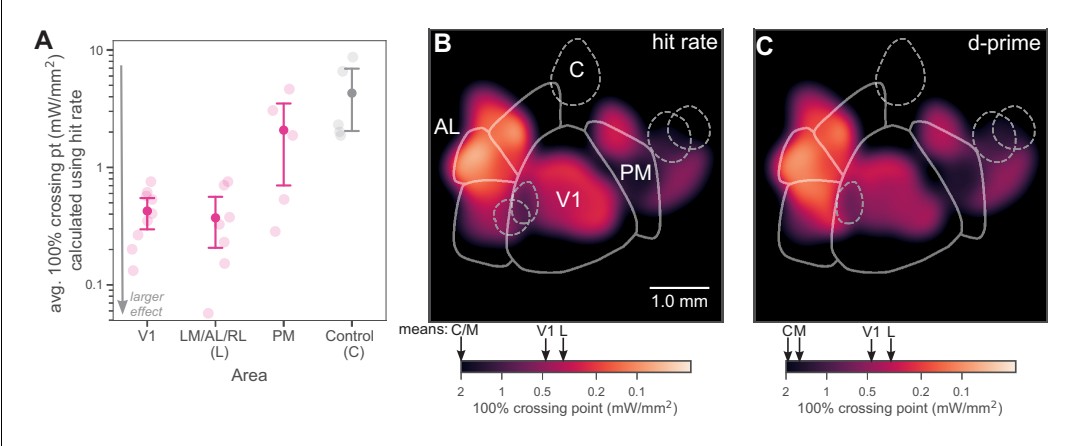

**Figure 5.** Inhibiting lateral areas produces larger effects than inhibiting PM. (**A**) Average 100% crossing point (± bootstrap 95% confidence interval) across all spots within a given area (V1/Lateral/Medial/Control). V1 and lateral areas (LM/AL/RL) require low light intensities to double threshold (mean 0.46, bootstrap 95% CI 0.33–0.58 mW/mm² and 0.36, CI 0.16–0.59 mW/mm², respectively). Inhibited areas within PM required much higher intensities (mean 1.7, bootstrap 95% CI 0.67–2.7 mW/mm²) to achieve the same effect, and control areas required even higher intensities (mean 5.1, bootstrap 95% CI 2.3–8.0 mW/mm²). (**B**) Heatmap of all inhibited spots, colored by hit-rate 100% crossing point. Light intensity required to produce effects in lateral secondary areas is similar to, or larger than, that needed in V1. Mean crossing points from spots within V1, LM/AL, and RL (**L**), PM (**M**), and control (**C**) are represented on color bar with black arrows. (**C**) Heatmap of all inhibited spots, colored by $d'$ 100% crossing point. When 100% crossing point is calculated in terms of $d'$, spots in V1 and lateral areas still require less intensity to double threshold (means 0.68 and 0.48 mW/mm², respectively) than spots in PM and control areas (means 3.0 and 6.1 mW/mm², respectively). Mean crossing points, color bar: same notation as in (**B**). The online version of this article includes the following figure supplement(s) for figure 5:

**Figure supplement 1.** $d'$ 100% crossing point calculations.

**Figure supplement 2.** Secondary visual area effects are not motor effects.

figure supplement 4). Given the small effect on false alarm hazard rate, which also varied little between areas, we expected little change in the pattern of our results when measured in terms of sensitivity. Indeed, calculating 100% crossing points in terms of sensitivity ($d'$) had no effect on the observed differences between areas (V1: Wilcoxon signed-rank test W = 15.0, $N_1 = N_2 = 9$, p = 0.37; LM/AL and RL: Wilcoxon signed-rank test W = 13.0, $N_1 = N_2 = 7$, p = 0.87; PM: Wilcoxon signed-rank test W = 7.0, $N_1 = N_2 = 5$, p = 0.89; Control: Wilcoxon signed-rank test W = 5.0, $N_1 = N_2 = 5$, p = 0.50) (**Figure 5C**, **Figure 5—figure supplement 1**).

We next examined changes in animals' response bias, or willingness to respond. There are several ways to characterize response bias, and we examined two different measures: absolute criterion and relative criterion (**Macmillan and Creelman, 2004**). Absolute criterion (c) is the distance of the decision threshold, in the stimulus space of signal detection theory, from the idealized zero point in that space. Absolute criterion varies with both hit rate and false alarm rate. When, as in our data, false alarm rate change is small compared to the change in hit rate, changes in absolute criterion are primarily due to changes in hit rate ($c = (z(H) + z(FA))/2$, where H and FA are the hit and false alarm rates, and $z(\cdot)$ is the inverse normal CDF). As expected, absolute criterion was changed by optogenetic suppression (data pooled across areas, regression of hit rate change vs absolute criterion change, N = 408 sessions, t = −26, p < 0.001). However, for this task, relative criterion ($c' = c/d'$) is a more appropriate measure of bias, as relative criterion accounts for changes in sensitivity. We saw little change in relative criterion with optogenetic inhibition (c' change, laser on–laser off: median −0.127, inter-quartile range 0.80, not significantly different from zero: linear regression of c' change on hit rate change t = -0.43, N = 408 sessions, p = 0.67). Thus, the systematic effects of optogenetic suppression on behavior are primarily due to decreases in sensitivity, not response bias; cortical suppression reduced animals' ability to detect the presence of the stimulus.

Effects of optogenetic inhibition could affect both sensory responses and, in principle, animals' motor responses. One way to determine whether our optogenetic inhibition affected motor responses is to vary the motor response the animal uses to report a stimulus change while keeping

the sensory stimulus the same. We did this by training an animal to perform the task using the fore-paw ipsilateral to the visual stimulus (in contrast to all other behavioral data in this work, for which animals were trained to use their contralateral forepaw). If inhibition affected motor responses, we would expect to see a different pattern of behavioral effects when the motor response type was changed. Instead, we saw similar effects in V1, LM/AL, and RL regardless of whether the ipsilateral or contralateral paw was used to report change (*Figure 5—figure supplement 2*). While these data cannot rule out some motor contribution, they are evidence that inhibition of these visual areas produces an effect primarily on sensory information, not motor execution.

Finally, the visual stimulus location we used for the experiments above (+25° azimuth, 0° elevation) is near the border of area PM, and it is possible that a stimulus in a more temporal location (i.e., displaced in the horizontal direction away from the animal's nose) could produce a larger neural response and perhaps reveal larger effects of inhibition. However, when we measured the effect of inhibition with a visual stimulus at a range of more temporal azimuths (+45–65° azimuth, 0° elevation), we found similarly small effects on behavior as with the more central stimulus (*Figure 4—figure supplement 1*). Thus, even across a range of retinotopic stimulus positions, the effect of PM inhibition on this perceptual behavior is not large.

In sum, inhibiting areas lateral to V1 (LM/AL and RL) produces the strongest effects on contrast-increment change detection, as large or larger than in V1. The fact that some secondary visual areas (the lateral areas) show these strong effects implies that V1 is not being directly decoded by remote areas, but instead that information must pass through these secondary visual areas before being used for contrast-change detection.

## Behavioral changes are primarily due to direct suppression, not feedback

Although we found above that inhibiting secondary visual areas impacted animals' contrast-detection behavior, it could have been possible that the effect of inhibiting secondary visual areas was merely to suppress V1 activity through feedback projections, similar to inhibiting V1 itself. To explore this possibility, we performed a series of electrophysiological experiments in which we inhibited in V1, lateral areas, or medial areas while simultaneously recording from neurons in the different areas.

We first examined how visual responses were suppressed at the site where optogenetic light was delivered (*Figure 6A*). We found that optogenetic stimulation of inhibitory neurons reduced or abolished visual responses at the site of light stimulation (units recorded from V1, LM/AL border or PM; *Figure 6B*; we targeted recordings to the superficial cortical layers, ≤500 μm from the cortical surface). LED intensities that produced behavioral effects (in V1, 0.43 mW/mm$^2$ to produce a 100% threshold change) completely suppressed neural activity directly under the LED (V1, bootstrap 95% CI 96–100% suppression of visual response without light, *Figure 6C*). Here, as throughout the physiological experiments, to obtain a large number of stimulus repetitions, we collected data while animals passively viewed the visual stimulus. To keep animals awake and alert, animals were water-scheduled and given drops of liquid during the recording period; the experiment was ended if they stopped licking to the reward (Materials and methods). We performed electrophysiological recordings with the same size stimuli used in the behavioral task, and used the same mapping protocols for behavioral and physiological experiments to identify areas. We used high-contrast stimuli for physiological recordings. Previous work (*Glickfeld et al., 2013a*) has shown that suppressing lower-contrast responses via optogenetic inhibition is similar to suppressing responses to high-contrast stimuli.

We next examined visual responses to contrast increments in the different visual cortical areas. Peak visual responses were similar in size between V1 and lateral areas, and both were significantly larger than peak responses in PM (*Figure 6—figure supplement 1A,B*). We measured visual stimulus latency (time to half-max response, logistic fit between $r_{min}$ to $r_{vis}$) in V1 to be 65 ms (bootstrap 95% CI 63–67 ms). There was a small delay in response in the lateral areas relative to V1 (time to half max response = 73 ms, bootstrap 95% CI 70–77 ms), but a notably larger delay for PM (time to half-max response = 90 ms, bootstrap 95% CI 87–93 ms, *Figure 6—figure supplement 1C*).

Since mouse V1 operates as an inhibition stabilized network (*Sanzeni et al., 2020*), at lower light intensities many inhibitory neurons *and* excitatory neurons should decrease their firing. However, at high light intensities some inhibitory neurons (especially the narrow-waveform inhibitory cells,

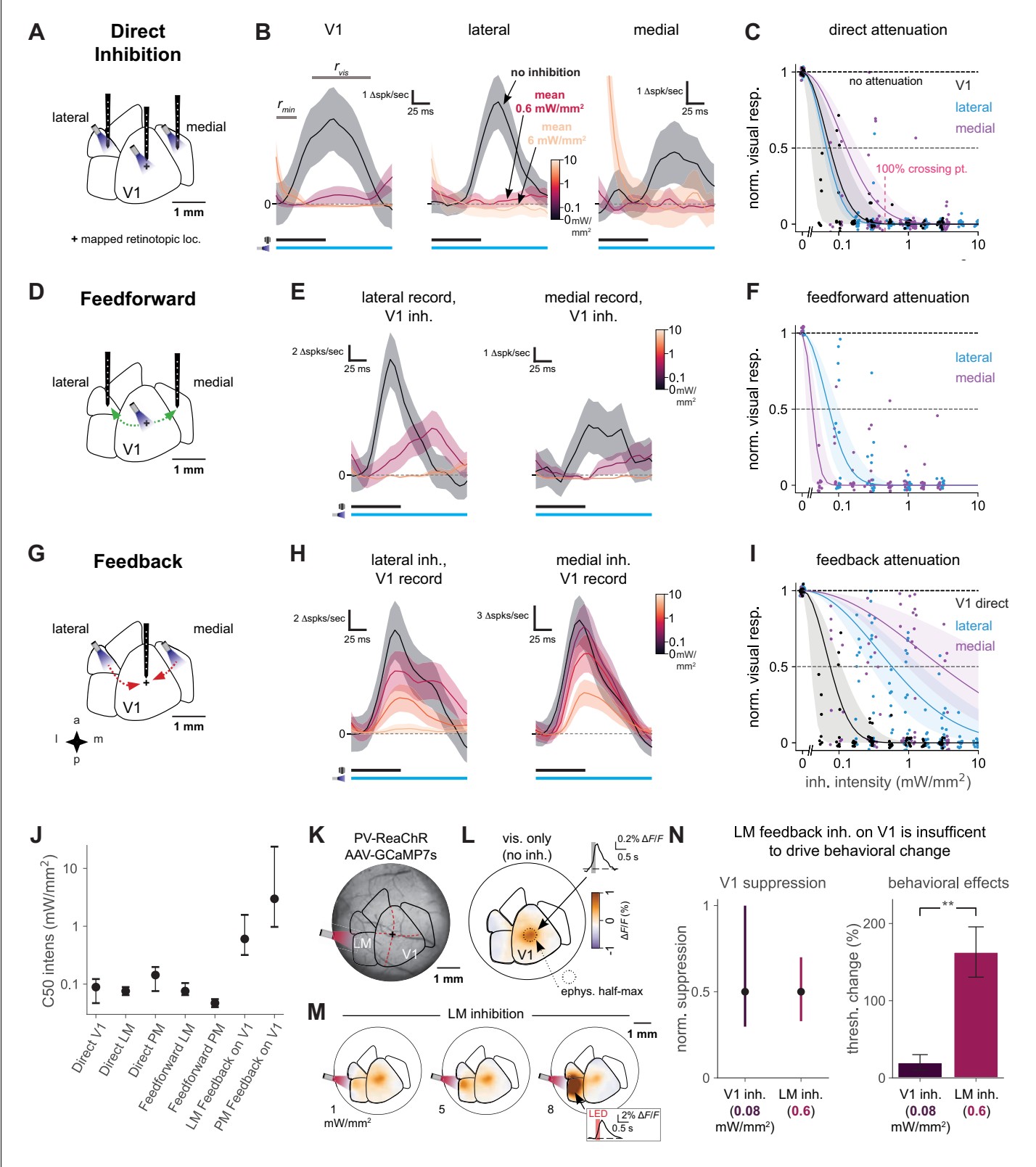

**Figure 6.** Direct activity reduction in secondary areas, not exclusively feedback suppression onto V1, accounts for degradation of behavioral performance. (A) Schematic of direct optogenetic inhibition of V1, lateral, and medial areas. Plus sign: visual stimulus at a mapped retinotopic location in V1. (B) Average visual responses in each cortical area with increasing amounts of optogenetic inhibition (0, 0.6, and 6.0 mW/mm²). Responses are

*Figure 6 continued on next page*

*Figure 6 continued*

subtracted by baseline $r_{\min}$, a 25 ms period during which optogenetic inhibition has reduced spontaneous neural activity but before visually evoked spikes arrive to the cortex. $r_{\mathrm{vis}}$: visual response period for analysis in **C**. Black and blue bars: duration of visual stimulus and optogenetic inhibitory stimulus, respectively. Vertical scale bars in panels vary, to more clearly illustrate relative suppression with optogenetic stimulation; quantification in panel **C**. The rightmost panel shows at the highest power a transient associated with ISN dynamics (*Sanzeni et al., 2020*, *Figure 6—figure supplement 3*), which has ended by the time the visual response analysis period begins. V1, lateral, medial panels: N = 14, 6, 7 single units. (**C**) Summary of direct inhibition effects on visually evoked responses for all intensities. One point is shown for every unit and every intensity level (four to five intensities per unit). Data set for A-I: 11 recording sessions in two animals; 22 electrode penetrations, each with eight recording sites, N = 79 total single units, see *Figure 6—figure supplement 3B* for electrode positions, see Materials and methods for selection of visually responsive units. Normalized visual response of 1 is no attenuation (black dotted line). Points are jittered slightly in both x and y directions for visual display, including at zero intensity where y = 1 for all points. Pink dotted line: Mean LED intensity for V1 100% behavior threshold change. Solid curves: Gaussian fits, shaded region: 95% CI via bootstrap. 50% suppression level shown by lighter dashed line. (**D**) Schematic of inhibition of feedforward V1 connections to secondary visual areas (green arrows). (**E**) Average visual responses measured in lateral areas and PM with increasing optogenetic inhibition applied to V1; conventions as in panel **B**. Lateral, medial panels: N = 4, 6 single units. (**F**) Summary of feedforward effects. Conventions as in panel **C**. (**G**) Schematic of feedback suppression of V1 during inhibition of secondary visual areas (red arrows). (**H**) Average visual responses measured in V1 with increasing optogenetic inhibition applied to lateral areas or PM. Lateral, medial panels: N = 26, 16 single units. (**I**) Summary of direct V1 inhibition versus feedback suppression for all intensities tested. Conventions as in panel **C**. (**J**) Intensities (of inhibitory optogenetic stimulus) that generate 50% visual response suppression for all methods of inhibition (direct, feedforward, and feedback) in all areas (V1, lateral areas, and PM). Error bars: 95% CI; all taken from intersection points with colored lines, shaded regions in panels **C**, **F**, and **I**. Higher intensities are required to produce suppression in V1 through feedback than through either direct or feedforward suppression of V1 to other areas. (**K**) Schematic of GCaMP7s imaging with LM inhibition in a mouse expressing ReaChR in all PV cells (PV-Cre;floxed-ReaChR mouse). (**L**) GCaMP7s response to flashed Gabor visual stimulus. Fluorescence map image is calculated by taking the frame-by-frame difference, approximating a spike-deconvolution filter; ΔF/F response without differencing is shown in inset. V1 activation (1.9% ΔF/F ± 0.22%, mean ± SEM) is restricted to the retinotopic location of the stimulus. V1 response is significantly greater than zero. (**M**) Responses to the visual stimulus paired with LM inhibition at three intensities. Increases in activity (orange) in LM/AL likely reflect increased firing of inhibitory neurons expressing ReaChR (inset: LM/AL light response time course has a decay consistent with GCaMP7s offset dynamics). V1 response was not significantly affected by LM inhibition (1.6% ΔF/F ± 0.16%, p for difference = 0.12, Wilcoxon U = 157.0, N1 = 30 trials, N2 = 30, one animal). (**N**) Intensity of optogenetic stimulation required to suppress V1 activity directly is much less than needed to achieve same suppression by illuminating LM. Intensities in both areas were chosen to produce the same mean suppression. Direct: 0.08 mW/mm$^2$, suppression mean 0.5, 95% CI 0.3–0.9, N = 4 recording sessions. Feedback: 0.6 mW/mm$^2$, suppression mean 0.5, 95% CI 0.4–0.7, N = 6 recording sessions. Behavioral effects at these powers are very different, indicating that behavioral effects of LM suppression arise principally via changing LM responses, not by feedback inhibition of V1 (V1 threshold increase at 0.08 mW/mm$^2$, N = 9 light spots; less than lateral threshold increase at 0.6 mW/mm$^2$, N = 7 spots; Mann–Whitney U = 3.0, one-sided, p < 0.0014). Errorbars in **B**, **E**, and **H**: SEM across trials of average across neurons on each trial.

The online version of this article includes the following figure supplement(s) for figure 6:

**Figure supplement 1.** Cortical responses to visual stimulus and response timing.

**Figure supplement 2.** Spatial effects in V1 of feedback vs. direct inhibition.

**Figure supplement 3.** Identification of inhibitory and excitatory units.

*Sanzeni et al., 2020*) increase their firing. To examine differences in baseline firing rate change due to stimulation, we sorted units by waveform width (*Figure 6—figure supplement 3A,B*), finding a bimodal distribution of spike waveforms (*Figure 6—figure supplement 3C*), as in *Sanzeni et al., 2020*. For simplicity, we excluded narrow-waveform units (likely the narrow-spiking basket cells, *Speed et al., 2019*; *Sanzeni et al., 2020*), to eliminate units that might increase their baseline rates to inhibitory stimulation. Thus, for analysis of visual responses below, we focused on wide-waveform units (excitatory and also likely some non-basket inhibitory cells). Also, we saw (*Figure 6—figure supplement 3D–L*) that secondary areas – lateral areas and PM – displayed signatures of inhibition-stabilized networks, supporting the idea that secondary visual cortical areas also operate in the inhibition-stabilized regime.

As V1 is a direct recipient of visual input to the cortex and sends projections to other visual areas (though secondary visual areas do receive projections from thalamic area LP, *Oh et al., 2014*), we expected that suppressing V1 would reduce responses in other visual areas by blocking the feedforward flow of visual information. Indeed, inhibiting V1 while recording downstream visual responses in either lateral areas or PM (*Figure 6D*) strongly decreased visual responses in both secondary areas (*Figure 6E*). We saw this effect at LED intensities comparable to those that produced 100% changes in behavioral threshold, and the feedforward suppression was even stronger at higher intensities

(*Figure 6F*). These measurements support the idea that much of the visual information seen in secondary visual areas passes through V1.

We next examined whether our behavioral effects in secondary areas originated solely from local suppression of visual responses or if they could have arisen from feedback suppression of V1. That is, we studied whether inhibiting secondary visual areas produced behavioral deficits by suppressing responses in the secondary area itself, or merely by acting on the V1 responses via feedback connections. To study this, we recorded in V1 and simultaneously inhibited lateral areas or PM while awake animals were passively viewing the visual stimulus (*Figure 6G*). We saw that optogenetic suppression of lateral areas did partially reduce V1 responses (*Figure 6H, left*). At light intensities that produced behavioral effects (0.37 mW/mm$^2$), inhibiting LM/AL produced moderate (39%, 95% CI: 24–57%) suppression in V1. However, the same light stimulation nearly eliminated visual responses in LM/AL (i.e., 100% suppression). Thus, the principal effect of stimulating LM/AL was to suppress LM/AL, with a smaller effect in V1 (*Figure 6C,I,J*). In contrast to the feedback effects on LM/AL, we found that suppressing medial areas had little to no feedback effect on V1 visual responses, even at the highest intensities (*Figure 6H, right*).

To characterize the spatial extent of feedback effects on V1, we used widefield calcium imaging. We expressed viral GCaMP7s in all neurons via multiple injections into a transgenic mouse expressing ReaChR-mCitrine in all PV interneurons (i.e., PV-Cre;ReaChR-mCitrine mouse; AAV-GCaMP7s injections; average green fluorescence due to GCaMP shown in *Figure 6K*). We imaged GCaMP responses with and without LM inhibition while the animal was shown the flashed Gabor stimulus (*Figure 6K*). As expected, GCaMP responses were observed at the V1 retinotopic location corresponding to the stimulus (*Figure 6L*).

When the visual stimulus was paired with LM inhibition, the average V1 response is unchanged (*Figure 6M*). (Slight reduction not statistically significant: without LM stimulation V1 response is 1.9% ΔF/F ± 0.22% [mean ± SEM], and with LM stimulation 1.6% ΔF/F ± 0.16%, p for difference = 0.12, Wilcoxon U = 157.0, N1 = 30 trials, N2 = 30, one animal. V1 response is significantly greater than zero, p < 1e-5, Wilcoxon U = 3.0, N = 30 stimuli. Signal is measured in ROI centered on V1 representation; fluorescence is measured in two frames or 200 ms after stimulus onset, relative to fluorescence in two frames before each stimulus onset). This result is consistent with the feedback suppression observed in physiology when LM was stimulated at high powers. At the site of optogenetic light stimulation in LM, we saw increases in fluorescence, with a decay time constant consistent with a GCaMP response (*Figure 6M*, inset; compare to V1 response decay, *L* inset), presumably due to activating inhibitory neurons at powers high enough to profoundly suppress excitatory cells, exiting the cortical ISN regime (*Sanzeni et al., 2020*; *Li et al., 2019* and *Figure 6—figure supplement 3*). These imaging data support the idea that the spatial pattern of V1 responses is not changed by LM suppression, and these data are consistent with our physiological measurements (*Figure 6I*).

Finally, to directly compare the effects on both areas (V1 and LM/AL) of optogenetically suppressing LM/AL, we examined V1 and LM/AL stimulation intensities that produced a matching magnitude of V1 suppression. Delivering 0.08 mW/mm$^2$ to V1 and 0.6 mW/mm$^2$ to LM/AL, while in both cases recording in V1, produced similar levels of suppression on the visual responses of V1 neurons (*Figure 6N, left*; these data were drawn from the Gaussian fits shown in *Figure 6C,I*). We were then able to examine in our behavioral data how the two types of stimulation (0.08 mW/mm$^2$ to V1; 0.6 mW/mm$^2$ to LM/AL), which affected V1 responses similarly, affected animals' behavior. LM/AL stimulation, and not V1 stimulation, produced large degradation of animals' contrast-change detection behavior (*Figure 6N, right*). We also compared direct and feedback V1 suppression across 0.4–0.8 mm spans of cortex, at both 0.1 and 0.3 mW/mm$^2$, and found that feedback suppression of V1 neural activity caused by LM direct inhibition was always two to five times lower than direct suppression of V1 neural activity (*Figure 6—figure supplement 2*).

Together, these observations argue that when lateral secondary areas LM/AL are suppressed, it is not the change in V1 responses that drives the behavioral effect, since stimulating V1 itself at an intensity necessary to produce the same V1 change has no effect on perception. Instead, it is likely the profound suppression of LM/AL activity (bootstrap 95% CI 99–100% suppression) at this light intensity that leads to the behavioral degradation. In sum, though inhibition of lateral areas does suppress activity in V1, feedback suppression alone could not explain the observed behavioral results.

## Discussion

This work examines the effect of inhibiting secondary visual areas in VGAT-ChR2-EFYP mice performing a contrast-increment change detection task. Neurons in V1, lateral secondary, and medial secondary visual areas change their firing rate in response to changes in stimulus contrast, but only suppressing either V1 or the lateral areas produced large effects on the behavior. Inhibiting the lateral areas sometimes caused deficits larger than those seen when inhibiting V1 directly. Behavioral effects were due to changes in sensitivity ($d'$) without large changes in response bias. When inhibiting the secondary areas, light intensities that produced behavioral changes produced the largest changes in neural firing in the secondary areas themselves, not in V1 via feedback suppression. Taken together, our results provide evidence that while substantial information about the contrast changes is present in V1, lateral secondary areas LM/AL and RL form an essential part of the neural circuit that controls this behavior.

### Behavioral changes are consistent with sensory, not motor, representations and are spatially specific

We performed a control experiment to determine whether inhibition affected sensory signals rather than motor planning or execution, and a second control to determine if our minimal PM effects could have been created by using a stimulus that was not well represented in the retinotopic map in PM. First, if the inhibition affected motor performance, we would have expected the effect to vary based on whether animals were using the contralateral or ipsilateral forepaw. Instead, we found that training animals to perform the task with their ipsilateral forepaw produced similar results as when animals used their contralateral forepaw (*Figure 5—figure supplement 2*). Second, we examined the possibility that our observed PM effects were weaker because PM may contain a weaker representation of the nasal than the temporal visual field. To rule this out, we measured the effect of inhibiting PM during the same task but performed with a stimulus further into the temporal visual field (45–65° azimuth relative to midline), where previous studies have found robust retinotopic representations in PM. Using this stimulus, we found similar or smaller magnitude effects as with our 25° eccentricity stimulus (*Figure 4*, *Figure 4—figure supplement 1*), arguing that the weaker effects in PM relative to lateral areas were not due to the location of the visual stimulus.

Our results examine the effect of inhibiting the cortex at a variety of different locations, and rely on several observations to establish spatial specificity. First, our key finding is that inhibiting lateral visual areas, LM and/or AL as well as RL, produces similarly sized, or larger, effects on behavior compared to inhibiting V1. Distance between the non-V1 spots and the V1 representation did not predict behavioral effect size (*Figure 3—figure supplement 1*), indicating the effects are due to inhibition at the site where the light was positioned. Second, we performed control experiments in which we moved the light spot between the V1 representation and LM/AL, within-subject, and found drastic falloff of behavioral effect relative to the V1 effect. These data indicate the degradation of perception we found by inhibiting lateral areas was due to a specific effect in LM/AL, and not merely due to spread across the cortex to the V1 representation. Third, the spatial specificity observed in behavioral experiments is supported by our neurophysiological recordings of cortical layer 2/3 neurons. We observed that the evoked visual response showed a spatial falloff consistent with, though slightly larger than, the light intensity patterns, or spots, that we used (*Figure 1L*). We recorded neural responses at several locations across V1, with recording sites generally in the superficial layers, which contain many cortico-cortical connections. These data confirmed that the spatial extent of suppression in V1 was similar when V1 was directly activated and when LM/AL was inhibited, though the direct suppression was significantly stronger across V1 than feedback suppression (*Figure 6* and *Figure 6—figure supplement 2*). Imaging V1 responses with L2/3-expressed GCaMP (*Figure 6*) also showed that the V1 representation was approximately the size of the area directly suppressed by our light spots, and that LM suppression affected an area of similar size. In sum, these characterizations of the spatial extent of suppressive effects support the idea that LM/AL suppression produces effects on behavior independent of V1.

A potential limitation of the electrophysiological and imaging results could be that they were recorded while animals passively viewed stimuli, outside the behavioral task. It is possible, for example, that decision behavior might modify the effect of optogenetic inhibition on V1 in one way and on secondary visual areas in a different way. One aspect of our recording approach is relevant here:

we gave animals rewards at regular intervals during recording and stopped the session if they stopped consuming the rewards (Materials and methods). Performing a rewarded behavioral task to high accuracy requires a high attentional or motivational state, and we reasoned that rewards would encourage high attention or alertness (and discourage a 'daydreaming' or low-arousal state, *McGinley et al., 2015*). Supporting the idea that our recorded data reflect similar physiological responses as during the task, several aspects of our behavioral effects also agree with our recorded neural data. These include the spatial falloff of the behavioral effects when light spots were moved, and also the fact that PM inhibition produces both weaker behavioral effects and weaker feedback effects onto V1 neurons than lateral area inhibition.

Another factor that may have changed our intensity thresholds, but did not affect the pattern of results across areas, is variability in effect size due to stimulation spot size. Even if there is some association of effect size with spot size, considering a subset of spots with similar sizes confirms our effects (e.g., *Figure 3D*, compare 'A', 'B', and the control spots shown in dashed lines; also see *Figure 4D*, regions marked 'B', 'C', etc). The idea that small variations in spot size do not lead to major differences in spatial spread is supported by a study of spatial spread of cortical optogenetic inhibition (*Li et al., 2019*), which found spread of inhibitory effects similar to ours (*Figure 1L*: mean radius of suppression = 0.73 mm; *Li et al., 2019*, their Figure 5H, radius of suppression 0.6–0.7 mm, estimated from their plot at minimum power, 0.5 mW), but with much smaller light spots (our spot FWHM 800 μm, Li et al. 400 μm at 4σ, or 240 μm FWHM). Finally, where we chose to site spots can also affect the heatmaps we use to display the spatial extent of behavioral effects (e.g., *Figure 5B, C*). While the heatmaps are useful to visualize average effects across areas, since they show effects only at the locations at which we sited stimulation spots, they do not capture the potential effect at every pixel location shown.

## Roles of mouse cortical visual areas in contrast perception behaviors

These findings show not only that lateral areas LM/AL and RL play a role in this behavior, but also that medial areas like PM do not play a major role. Both lateral and medial areas have previously been shown to encode different features of visual stimuli, such as spatial or temporal frequencies (*Andermann et al., 2011*; *Murakami et al., 2017*), and thus might play a role in object identification or localization, as in the 'what' and 'where' pathways of the primate visual system (*Mishkin and Ungerleider, 1982*). In this case, lateral and medial areas might be representing different features of the visual world, and though PM does not seem to be involved in simple visual perceptual tasks, other types of behavioral decision-making might expose a role for PM (see *Jin and Glickfeld, 2020*). While the role of PM in visually guided behaviors is currently unclear, it has distinct physiological responses from other visual areas. *Siegle et al., 2019* constructed a hierarchy of visual areas based on electrophysiological data (LM→RL→LP→AL→PM→AM) and predicted that medial areas (PM/AM) were higher areas within the hierarchy, which serve to amplify or decode change-related signals. Our data argue against the hypothesis that PM is downstream to V1 and lateral areas, at least in this task. If PM was important for processing signals for decisions about contrast changes, inhibiting medial locations should also serve to impair behavior, yet we found only weak mean effects. An important topic for future study is identifying the neural computations performed by PM that contribute to visual behavior.

In this work we examine several secondary areas: lateral areas LM, AL, and RL, and medial area PM. These areas have received consistent attention in past work, though several other secondary visual areas have been identified in different studies (*Garrett et al., 2014*; *Zhuang et al., 2017*). LM, AL, RL, and PM are the visual areas with the least positional variability relative to V1 (*Garrett et al., 2014*), that is, their location is most consistent across animals. While our methods might have, in principle, allowed us to differentiate LM and AL, AL is quite small and spatial spread of optogenetic suppression would make it difficult to completely affect one area and spare the other. Because of this, and because our main interest was understanding whether V1 alone was involved in this task or whether the task depended on at least one secondary visual area, to be conservative we describe these spots as being in LM/AL. Another medial area, AM, is found anterior to PM and is smaller in size than PM (*Garrett et al., 2014*; *Zhuang et al., 2017*), and some of our medial light spots may have reached this area, although the present data do not allow us to draw firm conclusions of the role of AM in this task. Finally, one of our control spots, the anterior-most spot in *Figure 3D*,

overlapped with the usual location of area A (*Zhuang et al., 2017*), and the lack of an effect at this spot suggests that area A is also not linked to performance of this task.

A recent study lends further support to the idea that PM is not crucial for contrast-perception or change detection tasks (*Jin and Glickfeld, 2020*). Jin and colleagues studied areas AL, LM, and PM and found, as we did, that lateral areas are involved in contrast-change detection tasks, and medial area (PM) inhibition has a smaller effect. Our electrophysiological and imaging measurements go beyond their behavioral results to show that the effect of inhibiting secondary visual areas is not caused by effects on V1, but rather is substantially caused by changes in the visual signals in secondary areas. Our behavioral results thus largely match theirs, but do differ in one way : on perceptual criterion. While *Jin and Glickfeld, 2020* found little or no change in sensitivity due to inhibiting PM, as we did, they found an effect on false alarm rates and criterion when PM was inhibited – and found shifts in false alarm rates for inhibiting PM both contra- and ipsilateral to the visual stimulus. The differences in false alarm rate change in our study and theirs may be due to the design of the behavioral task. We use a single light pulse train on all trials, shifted in phase, whereas *Jin and Glickfeld, 2020* used two types of trials, ON trials with light on throughout the trial, and OFF trials without optogenetic inhibition. Longer pulses could give more opportunity for false alarms to accumulate, amplifying any instantaneous (hazard) false alarm rate changes. Also, because subjects have some ability to choose their own criterion in perceptual behaviors (*Macmillan and Creelman, 2004*), differences in false alarm rates between the experiments might be explained not just by optogenetic differences between trials, but also potentially subjects' behavioral strategies and training regimen. In sum, while our study and this previous work show differences in false alarm rates, the findings are consistent in terms of sensitivity, or subjects' maximal ability to detect the stimulus. The lack of sensitivity change when PM is suppressed argues that PM is not part of the circuit that controls behavior in this contrast-perception task, while V1, LM/AL, and RL do play important roles in this behavior.

## Brain area interactions during perceptual behaviors

Our results shed light on the inter-area circuits used in a perceptual behavioral task. Recent work has shown that mouse premotor cortex is involved in downstream processing of sensory signals (*Wu et al., 2020*; *Zatka-Haas et al., 2020*), but it has been unclear whether secondary visual areas are involved in all or most sensory tasks that involve V1. Our data argue that lateral secondary areas are not bypassed during behavior and therefore suggest that secondary areas' activity is decoded into a choice or decision representation via those areas' projections to downstream regions. Evidence for choice or decision signals has been found in areas including frontal and premotor cortex and the striatum, but not in either V1 or secondary areas (*Steinmetz et al., 2019*), arguing that the decoding happens outside both V1 and the secondary areas. However, there are several caveats to this sequential interpretation of information flow from V1 to the secondary areas and then on to decoding areas. First, it is possible, though unlikely, that suppressing activity in cortical areas, as we did, acts on behavior by changing the activity of another area that is truly performing the neural computations for the task. For that to be true, it would require that the visual responses in each of these cortical visual areas are all epiphenomenal. A more likely alternative hypothesis to the sequential interpretation is that multiple brain areas are read out together in parallel. Deficits in visual detection similar to those we observed are also seen when the mouse SC is inhibited (*Wang et al., 2020*). Our observations and the SC results suggest that cortical circuits and the colliculus both carry information about visual tasks. Since in neither case does optogenetic inhibition totally abolish the behavior, the areas may each contribute information in perceptual tasks such as this. Moreover, in addition to parallel pathways involving the SC, secondary visual areas themselves may work partially in parallel to V1 (although our feedforward inhibition data show that input from V1 plays a major role in those areas' responses). The primary thalamic V1 feedforward source, the LGN, projects strongly to area PM and sparsely to area LM (*Bienkowski et al., 2019*). Regardless of how information is transmitted between areas, it is possible that when animals are performing near perceptual threshold, they use all available representations (e.g., multiple cortical areas, the SC, and potentially others) together to perform the behavior. If it was true that animals do use, or decode, all available representations in sensory areas, this would have implications not just for the circuits used during perceptual tasks, but also for the total amount of information available to the brain about particular stimuli, as calculations of this sort have previously focused only on V1 (*Kafashan et al., 2021*; *Rumyantsev et al., 2020*; *Stringer et al., 2019*).

How patterns of neural activity are processed across multiple brain regions to control behavior is central to our understanding of brain function. The role of secondary visual areas in sensory behaviors shown by these results suggests that the mouse visual system, like the visual systems of other species, contains a set of cortical areas that work in concert during perceptual behavior.

## Materials and methods

### Key resources table

| Reagent type (species) or resource | Designation | Source or reference | Identifiers | Additional information |
|---|---|---|---|---|
| Chemical compound, drug | Tamoxifen | Sigma-Aldrich | T5648-5G | |
| Genetic reagent (*M. musculus*) | VGAT-ChR2-EYFP | The Jackson Laboratory | RRID:IMSR_JAX:014548 | |
| Genetic reagent (*M. musculus*) | PV-IRES-Cre | The Jackson Laboratory | RRID:IMSR_JAX:008069 | |
| Genetic reagent (*M. musculus*) | ReaChR-mCitrine | The Jackson Laboratory | RRID:IMSR_JAX:024846 | |
| Genetic reagent (*M. musculus*) | Ai148 | The Jackson Laboratory | RRID:IMSR_JAX:030328 | |
| Genetic reagent (*M. musculus*) | Cux2-CreERT2 | MMRRC | RRID:MMRRC_032779-MU | |
| Recombinant DNA reagent | AAV9-hSyn-jGCaMP7s | Addgene | RRID:Addgene_104487 | |
| Software | MWorks | The MWorks Project | | mworks.github.io |
| Other | PEDOT;Poly (3,4-ethylenedioxythiophene) | Sigma-Aldrich | 687553 | |
| Other | PSS;Poly (styrenesulfonate) | Sigma-Aldrich | 243051 | |
| Other | C and B Metabond | Parkell | S380 | |
| Other | Kwik-sil | World Precision Instruments | KWIK-SIL | |

### Animals

All experimental procedures were approved by the NIH Institutional Animal Care and Use Committee (IACUC) and complied with Public Health Service policy on the humane care and use of laboratory animals. VGAT-ChR2-EYFP mice (ChR2 targeted at the *Slc32a1* locus, *Zhao et al., 2011*, Jax stock no. 014548, N = 16) were used for behavioral and electrophysiology experiments. Tamoxifen-treated Ai148;Cux2-CreERT2 mice (N = 3, GCaMP6f targeted at the *Igs7* locus and CreERT2 targeted at the *Cux2* locus, respectively, *Daigle et al., 2018*; Jax stock No. 030328, MRRC stock no. 032779-MU) were used for imaging experiments in *Figure 1—figure supplement 1*. A PV-Cre; ReaChR-mCitrine mouse (Cre targeted at the *Pvalb* locus and ReaChR targeted at the *Gt(ROSA) 26Sor* locus; *Lin et al., 2013*; Jax stock nos. 008069 and 024846, respectively) was used for LM inhibition during widefield GCaMP imaging in *Figure 6*. Nine females and eleven males were used in total, singly housed on a reverse light/dark cycle.

### Cranial window implantation and viral injection

Mice were given intraperitoneal dexamethasone (3.2 mg/kg) and anesthetized with isoflurane (1.0–3.0% in 100% $O_2$ at 1 L/min). Using aseptic technique, a titanium headpost was affixed using C and B Metabond (Parkell) and a 5 mm diameter craniotomy was made, centered over V1 (−3.1 mm ML, +1.5 mm AP from lambda). A 5 mm cranial window was cemented into the craniotomy, providing chronic access to visual cortex and surrounding secondary visual areas. Post-surgery, mice were given subcutaneous 72 hr slow-release buprenorphine (0.50 mg/kg) and recovered on a heating pad. For experiments involving widefield calcium imaging during LM inhibition, AAV9-hSyn-

jGCaMP7s was diluted 1:9 in sterile PBS and injected 300 μm below the surface of the brain. Multiple 800 nL injections were done at 200 nL/min to achieve widespread coverage across the 5 mm window.

## Hemodynamic imaging and visual area map fitting

To determine the location of V1 and surrounding visual areas, we delivered small visual stimuli to head-fixed animals at different retinotopic positions and measured hemodynamic-related changes in absorption by measuring reflected 530 nm light (*Schuett et al., 2002*). To find the locations of secondary areas, we characterized the position of V1 using responses to these stimuli, and using the V1 position, aligned a single map (first shown in *Figure 1A,B*) to each animal's cortex. (Mapping each secondary visual area using sign maps *Garrett et al., 2014* likely would have given more precise positioning of secondary areas, but due to spatial spread of optogenetic effects, such increased precision would have been unlikely to affect our results). We used hemodynamic intrinsic-signal imaging because the mouse line used for inhibition experiments expressed another fluorophore, YFP, that could have interfered with calcium indicator fluorescence. Imaging light was delivered with a 530 nm fiber-coupled LED (M350F2, Thorlabs) passed through a 530 nm-center bandpass filter (FB530-10, Thorlabs). Images were collected on a Zeiss Discovery stereo microscope with a 1× widefield objective through a green long-pass emission filter onto a Retiga R3 CCD camera (QImaging, Inc, captured at 2 Hz with 4 × 4 binning). We presented upward-drifting square wave gratings (2 Hz, 0.1 cycles/degree) masked with a raised cosine window (10° diameter) at different retinotopic locations for 5 s with 10 s of mean luminance between each trial. Stimuli were presented in random order at six or twelve positions in the right monocular field of view. The hemodynamic response to each stimulus was calculated as the change in reflectance of the cortical surface between the baseline period and a response window starting 2–3 s after stimulus onset, corresponding to the previously reported time course of visually evoked hemodynamic responses in mouse V1 (*Heimel et al., 2007*). We created an average visual area map, based on data from *Zhuang et al., 2017* and *Garrett et al., 2014*, and fit it to the cortex based on the centroids of each stimulus' V1 hemodynamic response.

## Widefield GCaMP imaging

To validate the placement of the hemodynamic map fits, we applied our hemodynamic imaging of V1 and map fitting procedure to animals in which we also performed GCaMP imaging of V1 and secondary visual areas. Animals (Ai148;CuX2-CreERT2; given tamoxifen at P18-22) expressing GCaMP6f in L2/3 cortical excitatory cells were head fixed while a Gabor stimulus (same stimulus used in behavior: +25° azimuth, 0° elevation, 14° FWHM, spatial frequency 0.1 cycle/degree) was shown in their right monocular visual field. We recorded GCaMP fluorescence changes through the 5 mm cranial implant over the contralateral visual cortex. Green fluorescence images were captured using an eGFP filter, at an average frame rate of 10 Hz (one stimulus frame followed by 32 post-stimulus frames, 4 × 4 binning) for 150 stimulus presentations. A 10 μL water reward was given randomly (10% probability) to keep animals awake and alert.

For GCaMP imaging with PV-ReaChR inactivation in LM/AL (*Figure 6*; AAV-GCaMP7s), we decomposed the imaging data using principal components analysis (PCA), discarded the first component, which encoded global fluctuation of the entire frame, and reconstructed the image stack. We then took a frame-by-frame difference ($F_{diff}$) at each pixel to approximate a deconvolution (*Zatka-Haas et al., 2020*). Image response maps show $F_{diff} - F_{diff0}/F_0$, where $F_{diff0}$ is $F_{diff}$ averaged over three frames (300 ms) just before each stimulus onset, and $F_0$ is average fluorescence over the same baseline frames. Insets in *Figure 6L,M*, to illustrate decay dynamics, show $\Delta F/F \triangleq (F - F_0)/F_0$, where $F$ is the pixel value without frame-by-frame differencing and $F_0$ is as above.

## Behavioral task

Mice were head-fixed in custom sound-attenuating boxes, and trained to hold a lever and release it when a Gabor stimulus increased its contrast relative to a gray screen (*Histed et al., 2012*). Mice were required to press a lever for a maximum of 4.1 s. At a random time point during this period, after a fixed period of 400 ms where the stimulus could not appear, a small Gabor patch (+25° azimuth, 0° elevation, 14° FWHM, spatial frequency 0.1 cycle/degree) was presented for 100 ms. Animals had 550 ms to report the stimulus by releasing the lever. Stimulus onset times were drawn from

a uniform or exponential distribution, adjusted manually to ensure short reaction times, and also to ensure no signatures of temporal guessing strategies (i.e. subjects releasing at a fixed trial time). Possible trial outcomes included a correct release within the reaction time window (hit), a failure to release the lever in response to the stimulus (miss), or a false alarm trial in which the animal released the lever before the stimulus appeared. Lever releases occurring in the first 100 ms following the stimulus presentation were counted as false alarms, as the animal could not have possibly perceived the stimulus and reacted in 100 ms. Trials resulting in hits were rewarded with 1.5–3 µL liquid reward. We began training with a full-screen, 100% contrast stimulus. As animals learned to perform in this task, the size and contrast of the stimulus were decreased and the location of the stimulus was slowly moved to the desired azimuth and elevation. As performance stabilized, we also increased the number of contrast levels to cover a range of contrasts, or difficulties, in order to determine the animal's psychometric threshold (the contrast at which the animal was getting 63% of trials correct). Sessions lasted until the animal reached its daily water supplement or until the animal stopped on its own, whichever came first (mean session length = 384 ± 97 trials). Because head-fixed mice make rare eye movements and SC inhibition does not systematically change those movements (*Wang et al., 2020*), mice have no visual streak and small difference in ganglion cell density across the retina, the secondary areas are small enough that our spots covered a large part of their retinotopic maps, and the few eye movements made by head-fixed mice are thought to be compensatory movements that occur with attempted head rotation (*Meyer et al., 2020*), we did not record animals' eye movements.

## Optogenetic inhibition

We used a fiber-coupled LED light source (M470F3, ThorLabs) to deliver illumination with a peak wavelength of 470 nm to the cortical surface. Fiber optic cannulae (400 µm core diameter, 425 µm outer diameter, 0.39 NA, Thorlabs CFMLC14L02) were cemented above specific cortical areas as determined by hemodynamic intrinsic imaging retinotopic maps. Light spot size and intensity contours were found by measuring the FWHM of the illuminated area (spot area mean ± SEM, 0.42 ± 0.13 mm$^2$, N = 34 spots; *Figure 1—figure supplement 2*). The area of the 50% contour was taken as the spot area for calculating light intensities, and the 50% contour was used as the boundary of each spot for plotting in figures. Once animals were performing well on the visual task and we had a full psychometric curve, we introduced a pulsed LED (on for 200 ms, off for 1000 ms) to each trial. The LED pulse train was randomly offset from the start of trial (uniform distribution) and pulsed 0–5 times before the visual stimulus was presented. On 50% of trials, the visual stimulus was presented 55 ms before the last LED pulse in the train (ON trials). On the remaining 50% of trials, the visual stimulus was presented 180 ms before the last LED pulse (OFF trials). Visual onset times were distributed according to a geometric distribution, whose mean was adjusted to balance the desire for a flat hazard function (to discourage waiting for later stimulus times; often reflected in false alarm rates <5% of trials) and to keep false alarms less than approximately 50% of trials. Once a visual onset time was randomly chosen, it was adjusted by lowering it to either 55 ms after the previous light pulse onset (ON trials) or 820 ms after the previous light pulse offset (OFF trials). Because of this, OFF trials were slightly longer than ON trials (ON: 1454 ms ± 187, mean ± SD trial length, OFF: 1942 ms ± 261 ms), though hazard rates for the two kinds of trials were nearly identical (*Figure 2*, *Figure 1—figure supplement 4*).

## Analysis of behavioral data

Analyses were performed with Python. Threshold increases were determined for each behavioral session by fitting a Weibull cumulative density function (hit rates) or Naka–Rushton function (d′; *Herrmann et al., 2012*) across contrast levels. The upper asymptote (lapse rate) and threshold (63% point between the upper and lower asymptotes, Weibull; 50% point, Naka–Rushton) were fit for each trial type with a single slope for LED ON and OFF trials. To exclude experimental sessions with insufficient data or where animals had poor motivation, only sessions with lapse rates <20% and sufficient trials at each contrast (average 10 trials per contrast level, per ON/OFF trial type) were included for further analysis (579 sessions total; 170 excluded, 409 sessions included). Only full sessions were excluded, and excluded data are not shown in any analyses in this work. To determine the 100% crossing point for each spot, or the intensity required to double the psychometric

threshold, a piecewise-linear function was fit to all percent increases obtained from various LED intensities. Fits were initially done on data with variable slopes. The slopes of spots with low slope CI ratios (<9 upper:lower CI, *Figure 1—figure supplement 3B*) were averaged and the regressions were re-fit using this determined slope (295.9). Piecewise-linear functions were fit in the same manner described above (*Figure 5—figure supplement 1*), first with variable slopes to V1 data and then again with a fixed average slope (261.2), in order to determine a $d'$ 100% crossing point for each spot. Heatmaps were generated by averaging the 100% crossing point at each pixel. Pixels with no data are black. False alarm hazard rates in *Figure 2C,D* were calculated by counting the number of occurrences in 1 ms bins, dividing by the number of trials that survived to that time, and multiplying by 1000 to give the hazard in units of percent probability of false alarm per second. Estimated false alarm hazard rates were calculated, for each session and ON/OFF block, by taking the total percentage of trials that ended in a false alarm, and normalizing by the average length of those trials in seconds.

## Electrophysiological recordings

For recording experiments, we first affixed a 3D printed ring to the cranial window to retain fluid. With the ring in place, we removed the cranial window and flushed the craniotomy site with sterile normal saline to remove debris. Kwik-Sil silicone adhesive (World Precision Instruments) was used to seal the craniotomy between recording sessions. Optogenetic light was delivered using the same optical fiber and cannula used for behavioral experiments. Cannula distance from the dura was adjusted, so light spots had approximately the same size as in behavioral experiments (average area $0.39 \pm 0.19$ mm$^2$; mean $\pm$ SD, N = 13 spots, measured with a macroscope, Opti-TekScope). We targeted a multisite silicon probe electrode (NeuroNexus; 32-site model 4 $\times$ 8-100-200-177; four shank, eight sites/shank; sites electrochemically coated with PEDOT:PSS [poly(3,4-ethylenedioxythiophene): poly(styrenesulfonate)]) to the center of the LED spot using a micromanipulator (MPC-200, Sutter Instruments). With the fiber optic cannula and electrode in place, we removed the saline buffer using a sterile absorbent triangle (Electron Microscopy Sciences, Inc) and allowed the dura to dry for 5 min. After insertion of the probes so that the deepest site was 500 μm below the dural surface, we waited 30–60 min without moving the probes to reduce slow drift and provide more stable recordings. We isolated single and multiunit threshold crossings (three times RMS noise) by amplifying the site signals filtered between 750 Hz and 7.5 Khz (Cerebus, Blackrock microsystems). The visual stimulus (90% contrast Gabor, 100 ms duration, same as during behavioral experiments) was presented on every trial, paired with a 200 ms LED pulse at varying intensity (180 reps each intensity). To keep animals awake and alert, animals were water-scheduled and a 1 μL water reward was randomly provided on 5% of the stimulus trials. Animals rarely stopped licking in response to reward (a sign of lowered alertness or engagement), but if they did cease to lick we terminated that experimental session.

## Electrophysiology analysis

Spike waveforms were extracted and sorted using Offline Sorter (Plexon, Inc). Single units were identified using waveform clusters that showed separation from noise and unimodal width distributions. Single units had SNR (*Kelly et al., 2007*; *Histed, 2018*; *Sanzeni et al., 2020*) of $3.5 \pm 0.36$ (mean $\pm$ SD, N = 171 units) and multiunit SNR was $3.1 \pm 0.28$ (mean $\pm$ SD, N = 54 units). Multi units were excluded from further analysis. In addition to examining the overall average spike rates of the responses, we calculated mean activity in three response windows: a baseline ($r_{base}$) window 40 ms in duration ending 10 ms prior to the delivery of the optogenetic stimulus, an inhibitory minimum ($r_{min}$) window of 25 ms starting 25 ms after optogenetic stimulation that ends just before spikes arrive to V1 (~40 ms, *Sanzeni et al., 2020*), and a maximum visual response ($r_{vis}$) window of 50 ms starting 65 ms post-visual stimulus for V1 and lateral areas. The max window was shifted 25 ms later for PM due to delay in the midpoint of the visual response (*Figure 1—figure supplement 1*, *Figure 6—figure supplement 1*). For analysis of visual response changes (*Figure 6*), we used the smaller subset of visually responsive V1, LM, and PM units, identified as having an $r_{vis}$ 10% or greater over $r_{base}$ with no inhibition (0 mW/mm$^2$ optogenetic stim). Response traces were baseline corrected to $r_{min}$, and the normalized change in spikes rates calculated as $r_{vis} - r_{min}/r_{vis}$ at 0 mW/mm$^2$.

## Acknowledgements

We are grateful to Victoria Scott for assistance with animal husbandry and to Patrick Wright for technical advice on hemodynamic imaging. We thank Richard Krauzlis, Bruno Averbeck, and members of the Histed lab for their comments on the manuscript.

## Additional information

### Funding

| Funder | Grant reference number | Author |
|---|---|---|
| National Institutes of Health | Intramural Program | Mark H Histed |
| National Institutes of Health | U19NS107464 | Mark H Histed |

The funders had no role in study design, data collection and interpretation, or the decision to submit the work for publication.

### Author contributions

Hannah C Goldbach, Bradley Akitake, Conceptualization, Data curation, Software, Formal analysis, Validation, Investigation, Visualization, Methodology, Writing - original draft, Writing - review and editing; Caitlin E Leedy, Conceptualization, Investigation; Mark H Histed, Conceptualization, Resources, Software, Formal analysis, Supervision, Funding acquisition, Validation, Investigation, Visualization, Methodology, Project administration, Writing - review and editing

### Author ORCIDs

Hannah C Goldbach (iD) https://orcid.org/0000-0002-5697-4694
Bradley Akitake (iD) https://orcid.org/0000-0002-1817-4573
Caitlin E Leedy (iD) http://orcid.org/0000-0001-9277-5409
Mark H Histed (iD) https://orcid.org/0000-0001-8235-7908

### Ethics

Animal experimentation: All procedures were conducted in accordance with the guidelines and regulations of the National Institutes of Health, according to an approved institutional animal care and use committee (IACUC) protocol (UNCB01) of the National Institute of Mental Health Intramural Program.

### Decision letter and Author response

Decision letter https://doi.org/10.7554/eLife.62156.sa1
Author response https://doi.org/10.7554/eLife.62156.sa2

## Additional files

### Supplementary files

• Transparent reporting form

### Data availability

Data with plotting code are available at: https://github.com/histedlab/code-GoldbachAkitake-visareas-simpleperception (copy is archived at https://archive.softwareheritage.org/swh:1:rev:7c8b71f1a52a76c19c0ba5c9e9cee023d8d04922/).

The following dataset was generated:

| Author(s) | Year | Dataset title | Dataset URL | Database and Identifier |
|---|---|---|---|---|
| Goldbach HC, Akitake B, Leedy | 2020 | Perceptual changes with optogenetic stimulation of visual | https://github.com/histedlab/code-GoldbachAki- | Github, code-GoldbachAkitake- |

| CE, Histed MH | areas of the mouse | take-visareas-simpleper-ception | visareas-simpleperception |
|---|---|---|---|

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
