## [Decision Letter]

**Acceptance summary:**

This well designed and carefully executed study addressed the contribution of secondary cortical areas to simple perceptual decisions in mice. The study of perceptual relevance of secondary visual areas fills an important gap in the rodent literature because of its prevailing focus on primary visual cortex. The authors used optogenetics to silence visual responses in secondary visual areas and assessed the behavioral and physiological effects of this manipulation during a contrast detection task. Despite the proximity between the primary and the secondary visual areas and the relatively limited spatial resolution of optogenetics, this technically challenging work provided compelling evidence for the importance of secondary visual areas for simple contrast detection. The work is likely to be highly relevant to investigators interested in sensory cortical circuits and their links to behavior.

**Decision letter after peer review:**

Thank you for submitting your article "Performance in even a simple perceptual task depends on mouse secondary visual areas" for consideration by *eLife*. Your article has been reviewed by two peer reviewers, and the evaluation has been overseen by a Reviewing Editor and Joshua Gold as the Senior Editor. The following individual involved in review of your submission has agreed to reveal their identity: Nicholas Steinmetz (Reviewer #1).

The reviewers have discussed the reviews with one another and the Reviewing Editor has drafted this decision to help you prepare a revised submission.

Summary:

The manuscript was well received by the reviewers, who felt that the work is timely and exciting and has a potential of being a valuable addition to the literature. However, they raise a number of substantive issues that must be addressed before the paper will be considered for publication in *eLife*. Both reviewers were concerned with the amount and strength of the data used to support the conclusions, with spatial specificity of the effect and with the way some of the data were presented and discussed. The main problems to be addressed in the revision are summarized below.

Essential revisions:

Data presentation

1) Please address the apparent paucity of data in the "test" experiment where stimulation was applied to the area between LM/AL and V1 to rule out distant effects in V1

2) Please provide information about the number of mice and experimental sessions that contributed to the data in Figures 5 C, F, I

3) "False Alarms" appear to be used interchangeably with other terms. Please select one and use it consistently.

4) The data in Figure 3—figure supplement 1B should include statistical analysis. Address the recommendation of reviewer 1 to replot the data for the two experiments separately, rather than averaging the two.

5) Please present absolute false alarm rates before and after stimulation and their distribution across the ITI.

6) If available, provide Information about layers of neurons recorded during neurophysiological experiments, to address the question is whether the output pathways from V1 are influenced by the inactivation of secondary visual areas. Discuss the relevance and the importance of layer specificity of the effects.

7) Provide the data 200ms before light onset and 200ms after light offset in Figures 5B,E,H

8) Provide statistical analysis in support of the statement that V1 response is reduced.

Spatial Spread of the effects

1) Explain how were the regional contours shown in Figure 1J determined. It is not clear how light delivery methods and locations were aligned across animals to generate maps of population results.

2) Explain how heatmaps of the behavioral effects were generated. Here, please address specifically the comments of reviewer 2 (points 2 and 3) and provide "greater characterization of the spatial spread across experiments (with aligned estimates across subjects) to rule out that the large effects in lateral areas are not due to combined inhibition of lateral areas plus the V1 representation of the stimulus location".

Behavioral issues

1) Please address the problem that stimulus conditions during electrophysiological recordings and behavioral testing were not equated. This requires that you explain and discuss the limitations in using activity recoded during a no-task conditions to infer activity underlying behavior.

2) Can you rule out a possibility that the observed behavioral effects are the result of a distraction rather than stimulation interfering with visual representation?

Other points to address in the manuscript

Alternative visual pathways, e.g. through the superior colliculus, relevant to the behavioral task should be considered

Reviewer #1:

This paper provides an answer to a question that has been difficult to answer: does inactivation of secondary visual areas impact perceptual reports differently than inactivation of primary visual cortex in the mouse? This is difficult because of the limited spatial precision of the method so far available (optogenetic inactivation via shining light on the surface of the brain), due to scattering and extensive lateral projections. The authors take a careful dive into this question and I believe they have the right set of experiments and controls to answer it (caveat that I don't think their behavioral task – a go/no-go detection task – is great for this purpose, see comment below). Therefore I think this paper will be a nice addition to the literature. My only main concern is with the strength/amount of data used to make the key arguments.

The key question for the behavioral experiments (Figures 1-4) is whether inactivation of LM/AL and RL degrades performance in a way that does not depend on directly modulating V1 activity. Thus this hinges on the test experiment "C1" in which a spot between LM/AL and the relevant part of V1 showed no effect. This is a great idea for a way to show that the LM/AL effects are not just from the wide spread of the opto inactivation into V1. But as far as I can tell this data point is literally one session with one mouse. I think we can't be confident about the conclusion on the basis of this one observation. In other words I don't feel yet convinced that the LM/AL effects aren't just because the neurons inactivated by the spots centered there also have distant effects in V1.

In Figure 5 C, F, and I – it needs to be stated how many mice and how many recording sessions went into the plot. If there is only one recording session (there might be, I can't tell) this seems to me like a big problem. If there are multiple recording sessions, I think it would be more appropriate to fit one curve per session and then average the curves across sessions, because within a session the responses of the neurons will be strongly correlated – in other words I think the error bars on the fit curves as currently shown are invalid due to these correlations.

Reviewer #2:

Goldbach et al. investigate the role of secondary visual areas (LM/AL, PM) for mice performing a contrast detection task. The authors conclude that the lateral visual areas (LM/AL) are as critical (if not more so) than V1 for behavior, while medial areas (PM) are less so. The topic is timely and exciting since higher visual areas have been largely understudied in comparison to V1 for mouse behavior. However, there are concerns with the spatial specificity of the methods, and inconsistencies across experiments provide difficulty in drawing clear inferences from the neural activity (out of task) to behavior. These aspects need to be addressed to provide greater confidence in the conclusions.

1) It is not clear how light delivery methods and locations were aligned across animals. The authors state “[v]isual area maps were fitted to the cortex based on centroid of each stimulus response”. However, the higher visual area locations, extents, and borders vary widely between mice (see Zhuang et al., 2017, Figure 3; Allen Brain Visual Coding White Paper, Figure 6, 9). These concerns apply for targeting and areal spread of light within subjects, but more so across subjects with "average" area contour plots (e.g., Figure 2,3). The white overlays shown throughout the figures give the impression of clear boundaries, but this is likely not the case, and requires greater care for population effects. This is critical for the main claim of the paper about the lateral areas-if these are also inhibiting V1 near the location of the stimulus (across the border with LM), the lateral area effects inherently include inactivating V1 (see point 2 below). The authors should ideally show visual sign maps/azimuth maps for individual mice (with alignment of the canulae to these individual maps), and then provide details for how they aligned areas and estimated light contours across mice to generate an aligned map of population results.

2) Measurement and description of the spatial spread of optogenetic inhibition requires greater clarity. It is unclear how the light profile was measured and if the light was adequately restricted to the target area within subjects. Authors do not provide enough detail to assess how beam profiles at the window / cortex were measured. It was not clear how heatmaps of the effect size were generated (e.g. Figure 1J, K), which are the main visualizations of spatial spread of behavioral effects. Further, the effects on neural activity should be measured and characterized within area (similar to Figure 5—figure supplement 3), not by inhibiting LM and recording the spatial effects in V1 (as presented in Figure 1L), as these likely depend upon the retinotopic alignment of the two sites (e.g., if LM prefers 20 degrees, while the V1 site prefers 60, the measurements may suffer from misaligned retinotopy rather than low spatial spread). If we go by the measures in Figure 5—figure supplement 3, the spatial spread is ~0.8 mm to 50% suppression, much larger than the measurement reported (0.42 mm^2^). Detailed measurements by Li et al., *eLife* (Figure 8) show that the spatial spread of inhibition is ~1mm with many different optogenetic methods. 1 mm is well within the range of nearby LM-V1 sites that both respond to a stimulus at 25 degrees. Perhaps this explains why LM inhibition produced 24-57% V1 suppression. The authors should provide greater characterization of the spatial spread across experiments (with aligned estimates across subjects) to rule out that the large effects in lateral areas are not due to combined inhibition of lateral areas plus the V1 representation of the stimulus location.

3) Related to the above, RL effect size heatmaps are often overlapping significantly with AL, PM overlaps with what should be AM, and many of the effect size overlaps with V1, again making clear interpretation about the specific areas roles in behavior difficult.

4) Electrophysiology was not performed during behavioral sessions. Recordings can still provide value if the experimental conditions mimic the behavioral situation as closely as possible. But, key discrepancies appear to diminish the relevance of the recordings for the behavioral effects. For example, the stimuli used during recordings are not at the threshold contrast, but at 90% contrast (Figure 5). Detection of this contrast is unaffected by light (Figure 1C). Further, light levels in V1 that produce shifts of the psychometric function (Figure 1C) completely suppress V1 activity for the 90% contrast stimulus; how then is the animal performing correct trials at even lower contrasts? It is important to address these difficulties, at least with deeper analysis of neural activity matched to behavioral conditions, and with discussion of limitations in using non-task activity to infer activity underlying behavior.

[Editors' note: further revisions were suggested prior to acceptance, as described below.]

Thank you for submitting your article "Performance in even a simple perceptual task depends on mouse secondary visual areas" for consideration by *eLife*. Your article has been reviewed by two peer reviewers, and the evaluation has been overseen by a Reviewing Editor and Joshua Gold as the Senior Editor. The following individual involved in review of your submission has agreed to reveal their identity: Nicholas Steinmetz (Reviewer #1).

Summary:

The reviewers were largely satisfied with the latest revision. However, reviewer 2 made a number of additional comments and suggestions, which if addressed will further improve the manuscript and its impact.

In your revision please follow the recommendation of reviewer 2.

Reviewer #1:

Thanks to the authors for carefully and thoroughly addressing my comments.

Regarding the point about statistics in Figure 5 (now Figure 6) – the concern is that neurons within a recording are correlated and can't be treated as independent samples – I am not sure why the authors did not take my very simple suggestion to fit one curve per recording session and show these curves and/or quantify error across them. I am not certain that the authors' argument about using a different resampling strategy for their bootstrap method addresses the concern, but I am not an expert on bootstrap methods, so perhaps it does. Nevertheless, given that I now know there are 11 recording sessions going into these figures, I feel significantly less concerned about this possible confound, and I do not feel that any further analyses are needed.

The new figure showing that lapse rates do not change with stimulation is a particularly useful and important addition to the paper (to me, this result is surprising, so it's great to learn this), as is the new figure detailing the false alarm rates.

I have no further critiques.

Reviewer #2:

A few Discussion points require greater clarity, especially regarding methodological constraints and how these may relate to their conclusions. Discussing these more clearly will help readers position the current findings relative to those showing different results regarding HVAs and neural activity during behavior.

Recommendations for the authors:

1) The authors should further clarify how choices for mapping HVAs may contribute to variability, and impact the conclusions. In the Response document, the authors describe how HVAs were identified (via identification of retinotopic extent of V1, and superimposing a standard HVA contour). This is helpful information that has not been added to the main text (as far as I can tell). In text that was added, the authors argue for low positional variability from prior studies as justification for aligning via V1 (which is ok), but this text does not develop how mapping V1, vs mapping the HVAs and boundaries themselves, might affect the conclusions. The authors parenthetically state that they cannot clearly dissociate LM/AL/RL effects, but don't mention the same issue regarding AM/PM overlap (a concern raised in prior comment 3); the authors claim to have added this text but it is not in the cited line numbers. The text should clearly explain that HVAs were identified with a standard contour map that was aligned by estimating V1 (again, which is ok), but different from and likely less accurate than mapping these small and heterogeneous areas mouse by mouse. The authors should discuss why this might be a reason why contributions of specific lateral areas and specific medial areas cannot be resolved here (see also point 2). It is important to be clear about identifying why methods here show coarser (and sometimes conflicting) roles of HVAs versus other recent work (Jin et al., 2020 and Siegle et al., 2020). The factors influencing the main point of the current paper (lumped lateral and medial areal contributions, versus more distinct contributions in other studies) can be made clearer.

2) Related to the above, Figure 3—figure supplement 1 again shows the role of spot size on behavioral threshold effects. This figure is a good control for the concern about encroachment effects, but shows that larger spots result in threshold changes with lower laser power (F has nearly double the spot size in V1 and requires a lower laser power for 100% crossing than D). This bears mentioning in the Discussion of spot size's potential role for behavioral effects. This is another potential factor limiting the resolvability of specific areal contributions, particularly given that the spread of light may activate multiple HVAs if they are not always located at / bounded by the standard contour (point 1).

3) The authors do not discuss constraints in extrapolating neural measurements from outside of the task to those underlying behavior. Active behavior versus passive stimulus presentation drives different dynamics throughout V1 and HVAs (Siegle et al., 2020, Figure 4). Additionally, several recent studies show that task-relevant and irrelevant motor factors are major contributors to neural dynamics in mouse sensory areas. Such factors (among others) likely affect the “linear decodability” from cortical activity during a task, but these cannot be assessed here. This should be stated clearly as a limitation.

---

## [Author Response]

Essential revisions:Data presentation1) Please address the apparent paucity of data in the "test" experiment where stimulation was applied to the area between LM/AL and V1 to rule out distant effects in V1

We added details on the number of sessions included previously, added an additional animal, and added a new figure showing the effect of moving spots.

2) Please provide information about the number of mice and experimental sessions that contributed to the data in Figures 5 C, F, I

Added.

3) "False Alarms" appear to be used interchangeably with other terms. Please select one and use it consistently.

Done. “False alarm” is now used instead of “early response” throughout.

4) The data in Figure 3—figure supplement 1B should include statistical analysis. Address the recommendation of reviewer 1 to replot the data for the two experiments separately, rather than averaging the two.

Added statistics: regression through all points, which are now shown separately.

5) Please present absolute false alarm rates before and after stimulation and their distribution across the ITI.

We made significant additions to false alarm (FA) data. Added new Figure 2 showing false alarm hazard rates. Added information about false alarm rates in Figure 1. Added new Figure 1—figure supplement 3 showing false alarm hazard rates across blocks, intensities, and areas.

6) If available, provide Information about layers of neurons recorded during neurophysiological experiments, to address the question is whether the output pathways from V1 are influenced by the inactivation of secondary visual areas. Discuss the relevance and the importance of layer specificity of the effects.

We now address these issues in Results. Unfortunately, we cannot draw conclusions about layer specificity from these data, so we have not added layer information to the neurophysiological results.

7) Provide the data 200ms before light onset and 200ms after light offset in Figures 5B,E,H

In Figure 6—figure supplement 2.

8) Provide statistical analysis in support of the statement that V1 response is reduced.

Added statistical analysis.

Spatial Spread of the effects1) Explain how were the regional contours shown in Figure 1J determined. It is not clear how light delivery methods and locations were aligned across animals to generate maps of population results.

We added this explanation.

2) Explain how heatmaps of the behavioral effects were generated. Here, please address specifically the comments of reviewer 2 (points 2 and 3)and provide "greater characterization of the spatial spread across experiments (with aligned estimates across subjects) to rule out that the large effects in lateral areas are not due to combined inhibition of lateral areas plus the V1 representation of the stimulus location".

We added this explanation and added new behavioral data to rule out the possibility that lateral area effects are due to light directed at lateral areas acting on the V1 representation directly. We move a light spot within a single subject, varying position in a single visual area map. We also added Figure 3—figure supplement 1, in which we show that the distance between the secondary area spots and the V1 representation does not predict the size of the behavioral effect.

Behavioral issues1) Please address the problem that stimulus conditions during electrophysiological recordings and behavioral testing were not equated. This requires that you explain and discuss the limitations in using activity recoded during a no-task conditions to infer activity underlying behavior.

Added this to the Discussion.

2) Can you rule out a possibility that the observed behavioral effects are the result of a distraction rather than stimulation interfering with visual representation?

Yes, we added both slope and lapse rate analyses and now discuss these issues directly.

Other points to address in the manuscriptAlternative visual pathways, e.g. through the superior colliculus, relevant to the behavioral task should be considered

This is now addressed in the Discussion.

Reviewer #1:This paper provides an answer to a question that has been difficult to answer: does inactivation of secondary visual areas impact perceptual reports differently than inactivation of primary visual cortex in the mouse? This is difficult because of the limited spatial precision of the method so far available (optogenetic inactivation via shining light on the surface of the brain), due to scattering and extensive lateral projections. The authors take a careful dive into this question and I believe they have the right set of experiments and controls to answer it (caveat that I don't think their behavioral task – a go/no-go detection task – is great for this purpose, see comment below). Therefore I think this paper will be a nice addition to the literature. My only main concern is with the strength/amount of data used to make the key arguments.The key question for the behavioral experiments (Figures 1-4) is whether inactivation of LM/AL and RL degrades performance in a way that does not depend on directly modulating V1 activity. Thus this hinges on the test experiment "C1" in which a spot between LM/AL and the relevant part of V1 showed no effect. This is a great idea for a way to show that the LM/AL effects are not just from the wide spread of the opto inactivation into V1. But as far as I can tell this data point is literally one session with one mouse. I think we can't be confident about the conclusion on the basis of this one observation. In other words I don't feel yet convinced that the LM/AL effects aren't just because the neurons inactivated by the spots centered there also have distant effects in V1.

We agree it is key in the behavioral experiments that the optogenetic light over LM/AL and RL does not exert behavioral effects by directly modulating V1 activity. To specifically address your points:

First, the data cited is N=15 sessions from one mouse (15 data points shown in Figure 3C; was Figure 2C in prior submission). The sample size was not explicitly stated before, but is now included both in the Figure 3 legend, where the spot is first shown, and in the Results section where Figure 3 is discussed. We have also added the number of sessions to all legends describing single spot crossing points (Figures 1,3,4).

Second, we have supplemented these data in two ways: (1) We repeated this experiment in a second animal and found little effect. (2) We added a figure (Figure 3—figure supplement 1) with this new data and also the prior data, in which we show behavioral effects from the interstitial (middle) spot and from a V1 spot just medial to it. These data serve as an internal control, showing that offsetting the spot from V1 by a short distance dramatically reduces the behavioral effect.

Third, other observations show that the behavioral effects are not merely caused by direct modulation of V1 activity by light. Other light spot locations at similar distances from the V1 retinotopic location produce smaller effect or no effect at all. Figure 4 shows 9 light spots (N=139 total experimental sessions) that each produced smaller or no effect, and five of those spots are at similar distances across the cortex to V1 as the LM spots (current Figure 5B). We have also now added a panel to the new figure supplement (Figure 3—figure supplement 1, panel A) showing that distance from V1 does not predict the effect size.

Finally, while the behavioral data are the key observations here, it is worth noting that the imaging (Figure 6KM) and electrophysiological (Figure 6A-I; Figure 6—figure supplement 3) data also show that LM inhibition has smaller effects on V1 responses than does direct V1 inhibition, further supporting that LM inhibition does not create effects by directly suppressing V1.

In Figure 5 C, F, and I – it needs to be stated how many mice and how many recording sessions went into the plot. If there is only one recording session (there might be, I can't tell) this seems to me like a big problem. If there are multiple recording sessions, I think it would be more appropriate to fit one curve per session and then average the curves across sessions, because within a session the responses of the neurons will be strongly correlated – in other words I think the error bars on the fit curves as currently shown are invalid due to these correlations.

The results presented in former Figure 5 C, F, and I (current Figure 6) are from 11 recording sessions in two animals (22 electrode penetrations each with 8 recording sites, N=79 total single units). These numbers were reported in former Figure 5—figure supplement 2 (now Figure 6—figure supplement 2) along with the electrode placements. We now include these numbers in the figure legend for Figure 6C.

The y-axes in Figure 6 C, F, and I measure the normalized amount of visual suppression – that is, the fraction by which the visual stimulus response is reduced. We used the bootstrap to estimate confidence intervals (CIs) of the fit curve, choosing units at random with replacement from the set of all recorded units. If units were strongly correlated within-session this could in principle bias the CI estimates. However, if such correlations are positive as often seen in the cortex, this should only bias our CI estimates to be too large – that is, the bootstrap we calculated would be overly conservative. However, to check explicitly whether correlations within-session would affect the pooled bootstrap, we calculated a nested bootstrap, randomly drawing samples with replacement from within each session and holding the number of units within each session constant. We found little effect on the error bars (Author response image 1).

**Author response image 1. sa2fig1:** Comparison of 95% confidence intervals when bootstrap groups were drawn from a pool of all trials (top row) or drawn randomly in a nested way: within each individual session holding the unit number within-session constant (bottom row). Number of bootstrap repetitions in all cases is 10,000. We observed only small differences in error bar sizes among all three classes of inhibition: direct, feedforward, and feedback. Position of points changes from top to bottom row because we randomly jittered each point for display purposes, see Materials and methods.

Reviewer #2:Goldbach et al. investigate the role of secondary visual areas (LM/AL, PM) for mice performing a contrast detection task. The authors conclude that the lateral visual areas (LM/AL) are as critical (if not more so) than V1 for behavior, while medial areas (PM) are less so. The topic is timely and exciting since higher visual areas have been largely understudied in comparison to V1 for mouse behavior. However, there are concerns with the spatial specificity of the methods, and inconsistencies across experiments provide difficulty in drawing clear inferences from the neural activity (out of task) to behavior. These aspects need to be addressed to provide greater confidence in the conclusions.1) It is not clear how light delivery methods and locations were aligned across animals. The authors state “[v]isual area maps were fitted to the cortex based on centroid of each stimulus response”. However, the higher visual area locations, extents, and borders vary widely between mice (see Zhuang et al., 2017, Figure 3; Allen Brain Visual Coding White Paper, Figure 6, 9). These concerns apply for targeting and areal spread of light within subjects, but more so across subjects with "average" area contour plots (e.g., Figure 2,3). The white overlays shown throughout the figures give the impression of clear boundaries, but this is likely not the case, and requires greater care for population effects. This is critical for the main claim of the paper about the lateral areas-if these are also inhibiting V1 near the location of the stimulus (across the border with LM), the lateral area effects inherently include inactivating V1 (see point 2 below). The authors should ideally show visual sign maps/azimuth maps for individual mice (with alignment of the canulae to these individual maps), and then provide details for how they aligned areas and estimated light contours across mice to generate an aligned map of population results.

We have improved our description of how these maps were aligned, and added new data and analyses.

Zhuang et al. (2017) used GCaMP signals in transgenic animals to compute their sign maps. Because we performed experiments in transgenic animals that express YFP across the cortex, it was not possible to obtain GCaMP maps in the same animals that we inhibited. We chose the VGAT-ChR2 line because ChR2 has relatively fast onset and offset and is expressed in all inhibitory cells, not just a single inhibitory cell class. It is also well-characterized for the effects of optogenetic light on E and I cells, see Sanzeni et al. (2020), Li et al. (2019). Finally, in our hands this line yields stable implants and stable behavioral effects over months.

Because of the difficulties in using calcium imaging to identify each visual area, we have used intrinsic signal imaging (absorption-based hemodynamic imaging, see Materials and methods) to identify V1 in each animal. Garrett et al. (2014) also used intrinsic signal imaging to compute sign maps. They computed the variability of secondary areas (their Figure 4), and found that LM, AL, RL, PM (and AM) were the least-variable secondary areas, with centroid scatter less than ~25% of each area’s size. While Zhuang et al. (2017) did not compute an equivalent metric directly, in their Figure 3D they show the variance of the LM, AL and V1 area centers is low (see dark colors at center of each area). In this work, we concentrated on the most reliably-located areas: LM, AL, RL, and PM. We did not compute sign maps for all areas, but instead aligned the secondary area maps to V1 by characterizing the V1 representation in detail by measuring V1 responses to 6 or 12 small stimuli of varying retinotopic location (Figure 1—figure supplement 1). Jin and Glickfeld (2020) used a similar visual stimulus set and approach to map visual areas.

One way to confirm map alignment is to move the light spot around in the same animal with a fixed map. We have now done this and include new data (Figure 3—figure supplement 1; N=2 animals), in which we first measured V1 responses and then moved the light to a spot between V1 and LM. We found a large reduction in behavioral response. These new data support that our LM/AL light spots did not spread to or inactivate V1 to cause the behavioral effects.

We have also added details on how light contours were estimated (see below, response to point 2). We have elected to keep the white area contours, and have revised text in Discussion to clarify that our area boundaries may have some variability, as discussed in Garrett et al. (2014), while emphasizing our main conclusion that there exist secondary areas that can impact this perceptual behavior.

2) Measurement and description of the spatial spread of optogenetic inhibition requires greater clarity. It is unclear how the light profile was measured and if the light was adequately restricted to the target area within subjects. Authors do not provide enough detail to assess how beam profiles at the window / cortex were measured. It was not clear how heatmaps of the effect size were generated (e.g. Figure 1J, K), which are the main visualizations of spatial spread of behavioral effects. Further, the effects on neural activity should be measured and characterized within area (similar to Figure 5—figure supplement 3), not by inhibiting LM and recording the spatial effects in V1 (as presented in Figure 1L), as these likely depend upon the retinotopic alignment of the two sites (e.g., if LM prefers 20 degrees, while the V1 site prefers 60, the measurements may suffer from misaligned retinotopy rather than low spatial spread). If we go by the measures in Figure 5—figure supplement 3, the spatial spread is ~0.8 mm to 50% suppression, much larger than the measurement reported (0.42 mm^2^). Detailed measurements by Li et al., eLife (Figure 8) show that the spatial spread of inhibition is ~1mm with many different optogenetic methods. 1 mm is well within the range of nearby LM-V1 sites that both respond to a stimulus at 25 degrees. Perhaps this explains why LM inhibition produced 24-57% V1 suppression. The authors should provide greater characterization of the spatial spread across experiments (with aligned estimates across subjects) to rule out that the large effects in lateral areas are not due to combined inhibition of lateral areas plus the V1 representation of the stimulus location.

Effects in lateral areas are not due to combined inhibition of lateral areas plus the V1 representation: We have added new data (Figure 3—figure supplement 1). These data, described also above in response to point 1, show that the behavioral effect of inhibiting the V1 representation is specific – there is dramatic falloff of the behavioral effect when we move the light spot from the V1 representation to a spot between V1 and the lateral areas (N=2 animals). Moreover, as you suggest, we wish to be cautious with any claim that inhibition of HVAs produces strictly stronger effects than inhibition of V1. It is certainly plausible that the effects of inhibiting higher areas could be larger than V1 (and we discuss differences between our data and Jin and Glickfeld’s below). This could be, for example, because the visual representation is smaller in the lateral areas than in V1, and thus light spots might be more effective in higher visual areas. But to show this conclusively, we believe evidence that exceeds our present scope would be needed, perhaps from experiments inhibiting combinations of V1 and higher visual areas. We have revised the text (Results and Discussion) to say that effects of inhibiting higher visual areas are at least as big as those due to V1 inhibition.

Light profile calculation: We now note in Materials and methods that we calculated the 50% contour of the light spot measured with a widefield camera imaging the brain surface, and that we used the contour as the spot outline seen in the heatmaps. (When spot diameters or FWHM are given, they are the equivalent diameter of a circle with the same area as the 50% contour). We then filled each contour with a constant value corresponding to that spot’s effect size (100% crossing point) and averaged the effect size at each pixel with more than one spot. We plotted the corresponding average value as a heat map color, as shown in the color bar. Pixels with no overlapping spots are black. The light profiles from our fibers (0.39NA, 400µm diameter) are approximately Gaussian. To be conservative, and avoid substructure in the heatmaps due to the Gaussian peaks of the light spots, we found the 50% contour and then filled the contour with a constant value for the heatmaps. We have added a supplementary figure (Figure 1—figure supplement 2) to illustrate the contour calculations, and revised the Materials and methods and Results to better describe the beam profile measurements and the heatmap calculations.

Differences in reported spread of suppression: In the text and in Figure 1L, we recorded visual responses along a line of sites from PM to V1 while inhibiting PM and recording in V1. This configuration (compared to recording in V1 while inhibiting LM) produces the smallest feedback effects in V1, presumably due to lower feedback influence on V1 by PM. In Figure 5—figure supplement (original submission), we also report effects of recording from LM to V1 during inhibition of either LM or V1. l.626 (orig. sub.) describes not the spatial spread of inhibition, but the size of the light spots, which is smaller than the spatial spread. For the PM-V1 recordings shown in Figure 1L, the farthest electrode, which produces the least suppression (15% ± 1% suppression, mean ± SEM, N = 45 units; 1.2 mm distance) is at the V1 representation of the stimulus. That is, when inhibiting in PM and recording in V1, there is very little suppression, as reflected in the half-max shown in Figure 1L.

Relationship between our measurements and those of Li et al. *eLife* (2019): First, the spread of our inhibition effects is comparable to theirs. Our Figure 1L: mean radius of suppression = 0.73 mm; Li et al. 2019: radius of suppression 0.6 – 0.7 mm (from their Figure 5H, estimated from their plot at minimum power, 0.5 mW). Second, we might expect their measures to differ from ours because the light spots they used were much smaller. Their spots are also Gaussian but “400 µm at 4σ”, or 240 µm FWHM, while our spots have a FWHM of ~800 µm (Figure 1L). Li et al. also used higher powers to produce their effects (their Figure 8), and overall much higher overall intensities of optogenetic stimulation, as the area of their spots was significantly smaller (inside the 50% contour: 0.05 mm^2^, ours 0.55 mm^2^). Therefore, while we produced behavioral effects between 0.1 – 0.5 mW/mm2 (Figure 1I, Figure 5A), the lowest intensity measured in Li et al. (their Figure 5H) was ~11 mW/mm^2^. Indeed, for the last figure of their paper, to measure paradoxical effects of inhibition, they used larger spots (2 mm at 4σ, or 1.2 mm FWHM), though they did not measure the falloff of inhibitory effects for these spots. Thanks for suggesting we compare to Li et al. We have added text to Discussion to make the above points.

3) Related to the above, RL effect size heatmaps are often overlapping significantly with AL, PM overlaps with what should be AM, and many of the effect size overlaps with V1, again making clear interpretation about the specific areas roles in behavior difficult.

Overlap with V1: While some spots do overlap the V1 border (for example the LM light spots), they are farther from the V1 retinotopic representation of the stimulus. We have added new data (new Figure 3—figure supplement 1) in which we move the optogenetic light stimulation locations within the same animal to show that it is light targeted to the V1 retinotopic representation that produces the behavioral effects.

Overlap between HVAs: It is true that RL and AL partially overlap, and we do not wish to claim that we have completely differentiated these areas. We have added additional text to the Discussion to clarify this point. Finally, while AM is differentiated from PM in Garrett et al. (2014) and Zhuang et al. (2017), it has a limited retinotopic representation compared to PM. However, the one medial spot at which we did see a significant effect of inhibition was at the anterior edge of PM (Figure 4D-E), which might be explained by overlap with AM. Thank you for pointing this out. We now address this in the Discussion.

4) Electrophysiology was not performed during behavioral sessions. Recordings can still provide value if the experimental conditions mimic the behavioral situation as closely as possible. But, key discrepancies appear to diminish the relevance of the recordings for the behavioral effects. For example, the stimuli used during recordings are not at the threshold contrast, but at 90% contrast (Figure 5). Detection of this contrast is unaffected by light (Figure 1C). Further, light levels in V1 that produce shifts of the psychometric function (Figure 1C) completely suppress V1 activity for the 90% contrast stimulus; how then is the animal performing correct trials at even lower contrasts? It is important to address these difficulties, at least with deeper analysis of neural activity matched to behavioral conditions, and with discussion of limitations in using non-task activity to infer activity underlying behavior.

Though recordings were conducted outside of behavior, we matched the position of the electrodes to the areas and retinotopic locations that were being targeted in the behavioral experiments. Electrode insertions relative to the area map were presented in (original submission) Figure 5—figure supplement 2B (now Figure 6—figure supplement 2B). The behavioral threshold shift illustrated in Figure 1C is a single example at a fixed inhibition power (0.46 mW/mm^2^) in V1. That power corresponds to near-complete suppression of V1 activity. However, suppression is not totally complete, which might mean that animals can perform correct trials with not very many neurons or spikes. (Compare to Dalgleish et al., 2020; though we do not feel strongly enough about this conclusion to make this comparison in the manuscript). In both V1 and LM/AL, across recordings and animals, we consistently observed strong behavioral deficits across a fairly limited range of inhibition intensities (0.3 – 1 mW/mm^2^), roughly on the edge of complete suppression. As you note, the stimulus contrast during recording was 90%; we chose a high contrast to put an upper bound on effects, as inhibition powers that abolish responses to 90% contrast should also abolish responses at lower contrast. In Glickfeld (2013) we measured effects of inhibition at different contrasts and found that indeed suppression was consistent across contrasts. We have added text to Results to make these points.

[Editors' note: further revisions were suggested prior to acceptance, as described below.]

Reviewer #2:A few Discussion points require greater clarity, especially regarding methodological constraints and how these may relate to their conclusions. Discussing these more clearly will help readers position the current findings relative to those showing different results regarding HVAs and neural activity during behavior.Recommendations for the authors:1) The authors should further clarify how choices for mapping HVAs may contribute to variability, and impact the conclusions. In the Response document, the authors describe how HVAs were identified (via identification of retinotopic extent of V1, and superimposing a standard HVA contour). This is helpful information that has not been added to the main text (as far as I can tell). In text that was added, the authors argue for low positional variability from prior studies as justification for aligning via V1 (which is ok), but this text does not develop how mapping V1, vs mapping the HVAs and boundaries themselves, might affect the conclusions. The authors parenthetically state that they cannot clearly dissociate LM/AL/RL effects, but don't mention the same issue regarding AM/PM overlap (a concern raised in prior comment 3); the authors claim to have added this text but it is not in the cited line numbers. The text should clearly explain that HVAs were identified with a standard contour map that was aligned by estimating V1 (again, which is ok), but different from and likely less accurate than mapping these small and heterogeneous areas mouse by mouse. The authors should discuss why this might be a reason why contributions of specific lateral areas and specific medial areas cannot be resolved here (see also point 2). It is important to be clear about identifying why methods here show coarser (and sometimes conflicting) roles of HVAs versus other recent work (Jin et al., 2020 and Siegle et al., 2020). The factors influencing the main point of the current paper (lumped lateral and medial areal contributions, versus more distinct contributions in other studies) can be made clearer.

In the last revision, we added text to Results to explain that we identified areas by mapping V1, as well as text to Materials and methods that provided additional detail.

“We identified a series of retinotopic locations within V1 (Materials and methods, Figure 1, Figure 1—figure supplement 1), and based on the location of V1, we identified several secondary visual areas, areas AL, LM, RL, and PM.”

V1 was not mentioned explicitly here and we have now added “V1” to this sentence.

That said, to make the point about contour map fitting more explicit, we have added another summary sentence to Materials and methods:

“To find the locations of secondary areas, we characterized the position of V1 using responses to these stimuli, and using the V1 position, aligned a single map (first shown in Figure 1A,B) to each animal’s cortex.”

We also now note that mapping secondary visual areas directly is likely to be more precise than identifying their positions from V1 maps.

The reviewer brings up the effects expected in different visual areas and why we chose to lump together some areas (e.g. LM and AL). The reviewer’s comment indicated to us that we were unclear on this topic, and so we have added a new paragraph to Discussion to cover it clearly.

We now discuss area AM there as well.

We do not think our results conflict with Jin et al. (2020). Their findings in LM and AL, in a contrast task, are similar to each other and to our LM/AL findings. Their findings in PM are also similar to ours (except for the false alarm differences that we discuss in a paragraph in Discussion). They do not study AM. Siegle et al. (2019 BioRxiv manuscript, now published in Nature 2021) is methodologically quite different from our work, as they use correlational (imaging and physiological measurements), not causal manipulations. In the last revision we had added: “Jin and colleagues found, as we did, that lateral areas are involved in contrast-change detection tasks, and medial area inhibition has a smaller effect.” We now also note there that they studied areas LM, AL, and PM.

2) Related to the above, Figure 3—figure supplement 1 again shows the role of spot size on behavioral threshold effects. This figure is a good control for the concern about encroachment effects, but shows that larger spots result in threshold changes with lower laser power (F has nearly double the spot size in V1 and requires a lower laser power for 100% crossing than D). This bears mentioning in the Discussion of spot size's potential role for behavioral effects. This is another potential factor limiting the resolvability of specific areal contributions, particularly given that the spread of light may activate multiple HVAs if they are not always located at / bounded by the standard contour (point 1).

The example shown in Figure 3—figure supplement 1 does produce a larger effect for the larger of the two V1 spots. These are just two spots, not a population effect, and although both spots cover the V1 representation, the larger spot is also at a slightly different location relative to the V1 representation (in our standard contour maps). Because of these differences, we have noted this spot size effect in the legend, but not in the Discussion.

3) The authors do not discuss constraints in extrapolating neural measurements from outside of the task to those underlying behavior. Active behavior versus passive stimulus presentation drives different dynamics throughout V1 and HVAs (Siegle et al., 2020, Figure 4). Additionally, several recent studies show that task-relevant and irrelevant motor factors are major contributors to neural dynamics in mouse sensory areas. Such factors (among others) likely affect the “linear decodability” from cortical activity during a task, but these cannot be assessed here. This should be stated clearly as a limitation.

Yes, this is a potential limitation. We have added a paragraph to Discussion to discuss this topic and clearly state that recordings were done outside the context of the task. One point we make is that, in our passive presentation, we give animals rewards, and stop sessions if they do not lick. Siegle et al. (2021) Figure 4 uses 60 min sessions of passive stimulus presentation without water scheduling or reward. This could lead to differences in brain state in the two experiments.